# Rescue of conformational dynamics in enzyme catalysis by directed evolution

Renee Otten [1], Lin Liu[2], Lillian R. Kenner[2], Michael W. Clarkson[1,4], David Mavor[2], Dan S. Tawfik[3], Dorothee Kern[1] & James S. Fraser [2]

Rational design and directed evolution have proved to be successful approaches to increase catalytic efficiencies of both natural and artificial enzymes. Protein dynamics is recognized as important, but due to the inherent flexibility of biological macromolecules it is often difficult to distinguish which conformational changes are directly related to function. Here, we use directed evolution on an impaired mutant of the proline isomerase CypA and identify two second-shell mutations that partially restore its catalytic activity. We show both kinetically, using NMR spectroscopy, and structurally, by room-temperature X-ray crystallography, how local perturbations propagate through a large allosteric network to facilitate conformational dynamics. The increased catalysis selected for in the evolutionary screen is correlated with an accelerated interconversion between the two catalytically essential conformational sub-states, which are both captured in the high-resolution X-ray ensembles. Our data provide a glimpse of an evolutionary trajectory and show how subtle changes can fine-tune enzyme function.

[1] Howard Hughes Medical Institute and Department of Biochemistry, Brandeis University, 415 South Street, Waltham, MA 02454, USA. [2] Department of Bioengineering and Therapeutic Sciences, University of California, San Francisco, San Francisco, CA 94158, USA. [3] Department of Biomolecular Sciences, Weizmann Institute of Science, Rehovot 76100, Israel. [4] Present address: Biological Sciences West, The University of Arizona, 1041 E Lowell Street, Tucson, AZ 85721, USA. These authors contributed equally: Renee Otten, Lin Liu. Correspondence and requests for materials should be addressed to D.K. (email: dkern@brandeis.edu) or to J.S.F. (email: jfraser@fraserlab.com)

The importance of protein dynamics in enzyme function has been under extensive investigation by experimental and computational methods and has become more widely accepted[1–5]. However, because proteins are inherently flexible, assigning a direct functional role to specific conformational changes has proved challenging. For human peptidyl-prolyl *cis/trans* isomerase CypA, a combination of biophysical experimental techniques has elucidated general principles of the energy landscape during catalysis[6–8]. As evolutionary selection acts on function, a new challenge is to understand how evolution shapes these energy landscapes[9]. This challenge is best exemplified by the common implication of protein dynamics as speculative explanation for the impressive functional improvements achieved via directed evolution, to improve enzyme activity[10–12], where often only minimal structural changes are observed[10,13]. More recently, studies using experimental and computational approaches have been reported that investigate the role of protein motions in evolution[14–19]. Here, we experimentally characterize changes in the energy landscape that emerge from directed evolution of CypA for enhanced catalytic activity. We find a direct correspondence of increased protein dynamics and faster catalysis with rate constants mirroring the catalytic turnover numbers along an evolutionary trajectory.

To directly observe the changes in an enzyme's energy landscape upon directed evolution, we turned to a previously designed second-shell mutation, S99T, in CypA that had three effects: inverting the equilibrium between two states that are essential for catalysis, decreasing their interconversion rate, and causing a parallel reduction in catalysis[8]. Can we restore the catalytic function via directed evolution and discern how the acquired mutations compensate for the impaired conformational dynamics

of the S99T mutant at the molecular level? Here, we use directed evolution to identify mutations that rescue the catalytic activity of S99T CypA. We measure the effect of the mutations kinetically, using NMR, and structurally, using room-temperature X-ray crystallography and multiconformer modeling. The rescued variant displays increased conformational exchange between two catalytically essential sub-states that are revealed directly by the X-ray measurements.

## Results

**Directed evolution of S99T CypA increases catalytic activity.** To enable directed evolution on S99T CypA, first a 96-well plate screen was developed that reports on the enzymatic activity of CypA. Proline isomerase activity is difficult to screen because of its high thermal background rate $(2–9 \times 10^{-3}\,s^{-1})$[20,21]. In addition, there are several proline isomerases in *Escherichia coli*, which complicates screening in cell lysate. To overcome these limitations, we took advantage of the *Pseudomonas syringae* phytopathogenic protease AvrRpt2, which is activated by eukaryotic, but not prokaryotic, cyclophilin homologs[22]. We expressed a library of CypA S99T variants created by random mutagenesis tuned to 1–3 mutations per gene, added inactive AvrRpt2 to cell lysate and monitored the cleavage of an AvrRpt2 substrate[23] (Fig. 1a). Besides revertants to wild-type CypA (Ser99), our initial screen of ~1000 variants identified a variant (S99T/C115S) with increased activity (Fig. 1b). A second round of ~1500 variants in the background of S99T/C115S CypA identified an additional mutation (I97V) with a further increase in activity. Extensive efforts to further improve enzymatic activity by many more rounds of evolution were unsuccessful. Both gain-of-function

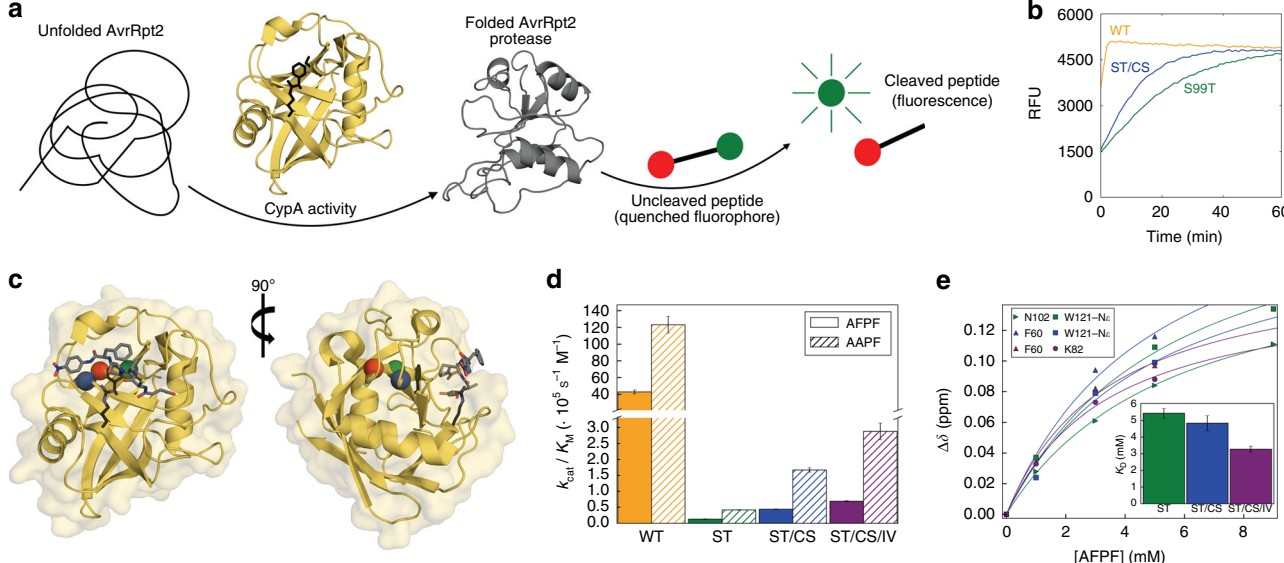

**Fig. 1** Directed evolution selects rescue mutations for catalysis. **a** Scheme of the assay used in directed evolution to identify CypA mutations with increased catalytic activity: CypA activity for folding of AvrRpt2 protease measured by AvrRpt2-mediated cleavage of the peptide Abz-IEAPAFGGWy-NH₂ (y = 3-nitro-Tyrosine). **b** Assay of directed evolution performed on cell lysate in 96-well plates to identify rescue mutations for S99T with increased CypA activity. Kinetics of peptide cleavage is shown for wild-type (yellow), S99T (green), and S99T/C115S (blue) CypA. **c**, **d** The severely catalytically compromised S99T mutant (green) is rescued by second-shell mutations (C115S, blue and I97V, red). **c** Sites of mutations are plotted onto the crystal structure (1RMH[68]) of CypA bound to Suc-AAPF-pNA (gray sticks) and the active-site residues Arg55 and Phe113 are shown in black stick representation. The overlay of NMR spectra shows that the overall structure of all CypA forms is very similar, with perturbations observed for residues close to the mutation site or in the dynamic network (Supplementary Fig. 1). **d** $k_{cat}/K_M$ values for wild-type, S99T, S99T/C115S, and S99T/C115S/I97V CypA measured by protease coupled hydrolysis[24] of Suc-AFPF-pNA and Suc-AAPF-pNA peptides (see Table 1). Error bars indicate the standard deviation obtained from triplicate measurements on at least three different enzyme concentrations. **e** $K_D$ values for the three mutant forms of CypA for Suc-AFPF-pNA measured by NMR chemical shift analysis from peptide titrations (see also Supplementary Fig. 2 and Table 1). Error bars denote the standard errors in the fitted parameters obtained from the global fit

**Table 1 Kinetic parameters and substrate affinity for wild-type CypA and mutants measured at 10 °C**

| CypA variant | $k_{cat}/K_M$ ($\cdot 10^5$ s$^{-1}$ M$^{-1}$) | | $K_D$ (mM) | |
| --- | --- | --- | --- | --- |
| | Suc-AFPF-pNA | Suc-AAPF-pNA | Suc-AFPF-pNA | Suc-AAPF-pNA |
| wild-type | 42.82 ± 2.38 | 123.22 ± 9.95 | | 1.8 ± 0.14[a] |
| S99T | 0.13 ± 0.01 | 0.42 ± 0.01 | 5.4 ± 0.5 | 6.7 ± 0.8[a] |
| S99T/C115S | 0.44 ± 0.01 | 1.67 ± 0.07 | 4.8 ± 0.4 | |
| S99T/C115S/I97V | 0.69 ± 0.02 | 2.88 ± 0.26 | 3.3 ± 0.2 | |
| C115S | | 53 ± 8 | | |
| I97V | | 122 ± 12 | | |
| I97V/C115S | | 88 ± 9 | | |

[a]Measured for Suc-AAPF-pNA at 6 °C as described earlier[8]

mutations are in proximity of Thr99, but not in direct contact with the peptide substrate (Fig. 1c).

To quantitatively characterize the mutants selected by directed evolution, catalytic efficiency was measured for purified proteins using the well-established coupled chymotrypsin assay[24]. This spectrophotometric assay is based on chymoptrysin's isomer-specificity for only *trans*-peptide bonds, and product formation (i.e., para-nitroanalide) is measured. Each mutation contributes additively to the increase in activity, measured as $k_{cat}/K_M$[24], and is consistent across two substrate peptides (Fig. 1d and Table 1). The intrinsic limitation of this enzymatic assay is the inability to measure at substrate-saturating concentration and, therefore, $k_{cat}$ and $K_M$ cannot be separated. In an effort to determine whether the observed changes in $k_{cat}/K_M$ are due to changes in turnover number or substrate affinity, substrate binding was detected directly by NMR titration experiments. The dissociation constants are only slightly changed relative to the S99T mutant (Fig. 1e, Supplementary Fig. 2 and Table 1) suggesting that the two mutations function by modulating the turnover rate rather than substrate binding. This finding is in agreement with the comparison between wild-type CypA and the S99T mutant, where the $K_D$ was only weakened by 3.7-fold, whereas $k_{cat}/K_M$ decreased 300-fold[8]. Our data show that the S99T/C115S and S99T/C115S/I97V mutations increase $k_{cat}/K_M$ relative to S99T by 3-fold and 5-fold, respectively. However, directed evolution in the context of the S99T mutation only restores about 2% of the full catalytic efficiency of the wild-type protein (Fig. 1d).

**NMR dynamics measurements of rescue mutants**. To determine whether the rescue in catalysis is due to faster protein motion, we performed NMR dynamics experiments on the S99T, S99T/C115S, and S99T/C115S/I97V mutants (Fig. 2, Supplementary Fig. 1 and Supplementary Data 1–6). Previous NMR CPMG dispersion experiments suggested a direct link between catalytic efficiency and the speed of a conformational change in a dynamic network, labeled group-I, for both wild-type (WT) and S99T CypA (Fig. 2a, red)[7,8]. In short, in S99T motions of the group-I residues became slow on the NMR timescale compared to fast motions observed in WT (a decrease of about two orders of magnitude). In contrast, a second dynamic process, group-II, comprised of loops adjacent to the active site (Fig. 2a, blue), is insensitive to mutation and displayed faster dynamics. Despite the lack of correlation between the dynamics and catalysis in the group-II residues in S99T, these residues have recently been proposed to be directly linked to catalysis in CypA[25].

To understand the changes in the energy landscape during the directed evolution that led to faster enzymes, we first turned to

well-established $^{15}$N-CMPG experiments. Interestingly, the exchange contribution, $R_{ex}$, for group-I residues gets bigger with consecutive rescue mutations (Fig. 2b and Supplementary Data 3), indicating that these mutations increase the slow interconversion rate from the major to minor state (i.e., $R_{ex} = k_{maj\rightarrow min}$). However, rate constants and populations are often not accurately determined by CPMG profiles for processes in the slow-exchange regime. We note that the fast loop motion of group-II (residues 65–80) is observed in all enzyme forms and remains essentially unaltered (Fig. 2a–c, e, g and Supplementary Data 1–3).

For a quantitative understanding of the mechanism underlying the increased catalysis along the directed evolution trajectory, we applied a powerful NMR method for studying systems in slow exchange, chemical exchange saturation transfer (CEST) spectroscopy[26]. The $^{15}$N-CEST experiments identified a large number of residues in slow exchange in all mutant forms of CypA (Fig. 2c–h and Supplementary Figs. 3–5 and Supplementary Data 4–6) and delivered two key results. First, for the rescue mutants we observe two distinct slow dynamic clusters that differ in kinetics and populations (Fig. 2e, g, Supplementary Figs. 4 and 5 and Table 2). The exchange profiles of 46 residues for the double-, and 55 residues for the triple-mutant were globally fit to a linear, three-state exchange model (see Methods). Second, we discover a gradual increase in the interconversion rate from the major to the minor conformation for the slow process centered around group-I residues ($k_{maj\rightarrow min}$ 2.3 ± 0.1 s$^{-1}$ for S99T, 8.2 ± 0.4 s$^{-1}$ for S99T/C115S and 11.5 ± 0.5 s$^{-1}$ for S99T/C115S/I97V, respectively).

NMR relaxation experiments enable us to "see" the prevalence of exchange on different timescales and to unravel their importance in biological processes[1]. The correspondence between $k_{maj\rightarrow min}$ and $k_{cat}/K_M$ (Fig. 3a) corroborates our hypothesis that the increased dynamics of group-I residues is indeed responsible for the rescue in enzymatic activity. The good correlation between the chemical shift differences, $\Delta\delta_{AB}$ and $\Delta\delta_{AC}$, in the two rescue mutants (Fig. 3b) indicates that the exchange processes are the same by nature. The newly identified second slow-exchange process is located on the opposite side of the protein and partially overlaps with the faster loop motion detected by CPMG experiments (Fig. 2e, g). The rather large chemical shift differences observed for this exchange process (Fig. 3b, green squares) agree well with the predicted values of going to a more extended/unfolded state (Supplementary Fig. 6) and are, therefore, unlikely to be relevant for catalysis.

In wild-type CypA, overall turnover is dictated by the interconversion between the major and minor conformations as the dynamics measured on the substrate-bound enzyme coincides with the rate of *cis*-to-*trans* isomerization of the bound substrate[7,20]. This suggests that the two distinct protein conformations are needed to accommodate the *trans* and *cis* form of the substrate. Secondly, the conformational exchange occurs on similar timescales in both the apo enzyme and during turnover[7]. To confirm that such a correspondence between the measured dynamics in the apo and turnover protein holds true for the mutants, CPMG and CEST experiments were performed on S99T during enzymatic turnover of a substrate peptide. These experiments on the mutants proved to be difficult and only possible for S99T due to stability, and the weak affinity for the peptides allowed for a maximum of ~70% saturation. For S99T during catalysis, we indeed observe both fast loop movement, and slow conformational dynamics in the group-I residues, very similar to the apo protein (Fig. 3c–e and Table 2). We conjecture that the increase in $k_{maj\rightarrow min}$ in the rescue mutants is responsible for the increase in $k_{cat}/K_M$. Note that interconversion in the apo state ("recycling" of the enzyme from the *trans*- to the *cis*-binding

conformation in the chymotrypsin assay) and/or in the substrate-bound form can be rate-limiting. In summary, these data show that the intrinsic dynamics in the group-I residues are rate-limiting for the catalytic cycle, as the severe reduction of the protein's interconversion rate parallels the reduction in $k_{cat}/K_M$ (see Tables 1 and 2)[7,8].

**Structural basis of increased dynamics in rescue mutants**. To reveal the structural basis of the increased dynamics in the group-I residues and hence increased catalysis along the evolutionary trajectory, we collected room-temperature X-ray data for S99T/C115S/I97V (Fig. 4). Alternate conformations were identified using qFit[27] and the final multiconformer model was obtained after subsequent manual adjustments and occupancy refinement (Supplementary Fig. 7). In S99T/C115S/I97V the Phe113-in to

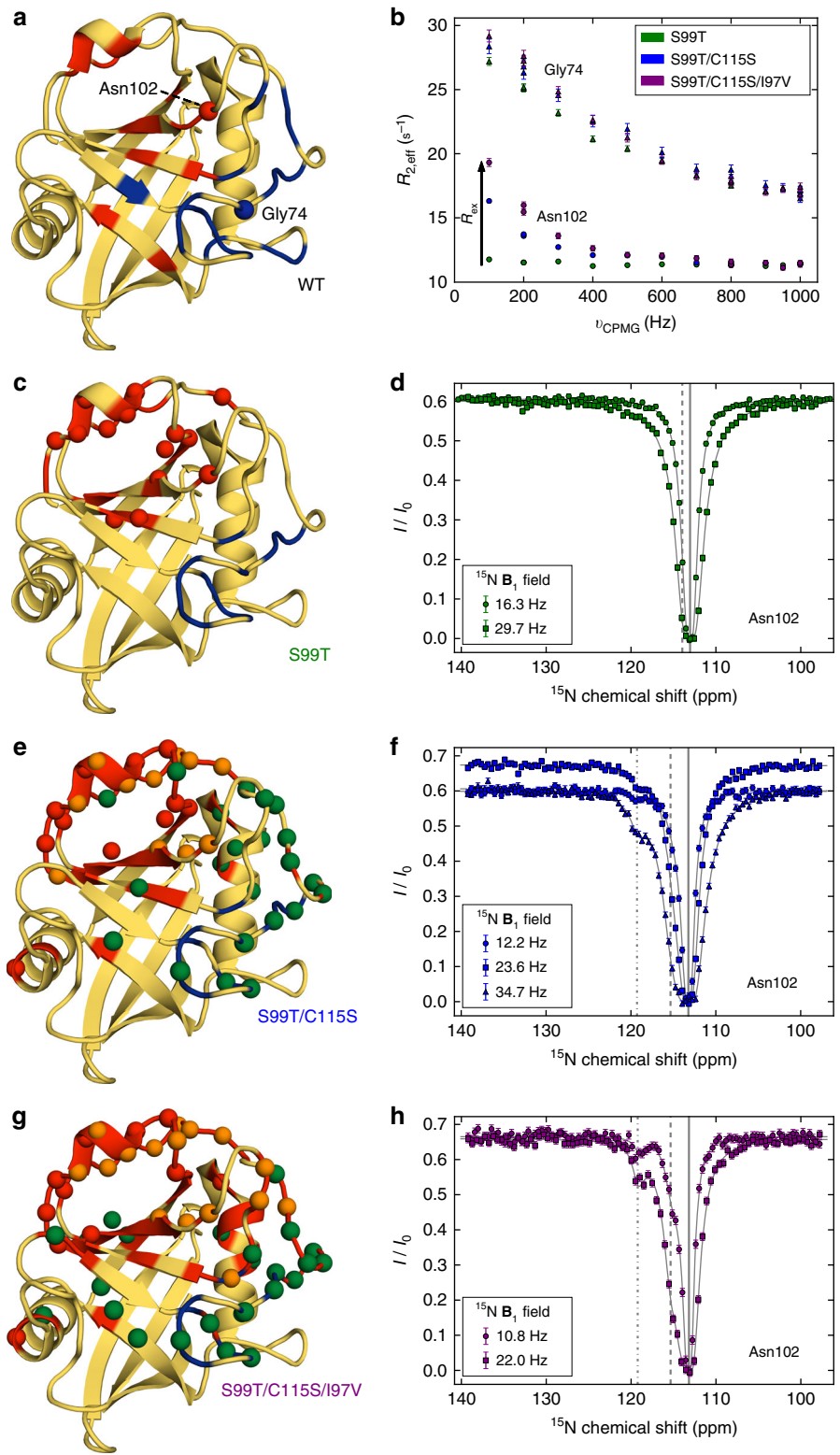

Phe113-out conformations have refined occupancies of ~20 and 80%, which is intermediate between the 65 and 35% ratio observed for WT (3K0N)[8] and the 100% occupancy of the Phe113-out conformation observed for S99T (3K0O)[8] and S99T/C115S (6BTA, Supplementary Table 2). Although we hypothesize, based on the NMR data, that the Phe113-in conformation is present at low occupancy in both S99T and S99T/C115S, it is not observable in the crystallography data. A swap of the major/minor states from WT to S99T/C115S/I97V is observed for the group-I residues, thereby directly delivering the atomic structures of the conformations for which we measured their interconversion rate by NMR. Both the C115S and I97V mutations subtly reduce the amino acid size, and combined partially restore the "Phe113-in" conformation as a minor conformation that can now be directly observed in the electron density (Fig. 4b, d–f). The interpretation of increased conformational heterogeneity of Phe113 and Thr99 was additionally confirmed by an alternative ensemble refinement method (Supplementary Fig. 8). The size reduction by the C115S mutation could contribute to the faster transition between the major and minor conformation due to the relief of a clash between the larger sulfur atom and Thr99 (Fig. 4e).

## Discussion

The connection between these alternate conformations and the collective fitting of the NMR dynamics is further buttressed by analysis of contacts between alternative conformations[28], which identifies a network across the protein that coincides with the group-I residues (Fig. 4g). Although the detection of alternate conformations by X-ray diffraction does not per se deliver information about correlated motions, a detailed analysis of the steric constraints of group-I residues (Fig. 4d–f) exposes how motions from the active site are propagated in a correlated manner. These structural results, together with the characterization of collective dynamics by NMR, reveal how the mutations selected by directed evolution have rewired the internal packing to increase the dynamics of Phe113, which likely makes distinct contacts with the cis- and trans-substrate (Supplementary Fig. 9), correlated with surrounding group-I residues during the catalytic cycle.

Why did additional rounds of directed evolution fail to yield further improvement in the catalytic rate? We speculate that the enzyme may be in a local minimum in the fitness landscape, and that a specific combination of mutations is needed for further improvement, including mutations that are neutral or mildly deleterious mutations on their own, in agreement with the dominance of epistasis in protein evolution[29]. There is a weak epistatic relationship for C115S: it decreases activity in the wild-type background and increases activity in the S99T background (Table 1). Alternatively, any further catalytic improvement via additional mutations may be precluded in the context of the severe S99T mutation.

Characterizing how directed evolution shaped the energy landscape for enhanced catalytic activity by solely increasing the conformational interconversion rates in a specific dynamic network has broad implications for resolving controversies about the role of protein dynamics in enzyme catalysis of modern enzymes, and in discovering the mechanism of improved catalysis via directed evolution. We briefly discuss both points in respect to pertinent current views in these fields. Fueled by a strong dispute about protein dynamics impact on catalysis[30,31], the field has focused on characterizing several enzymes mechanistically in great detail using a combination of experimental and computational approaches, including CypA[6–8,25,32]. A two-state ensemble calculation using exact NOEs as constraints revealed an open- and closed-state of free CypA with respect to the position of the

**Table 2 Exchange parameters extracted from $^{15}$N-CEST experiments for CypA variants measured at 10 °C**

| CypA variant | Exchange parameters | | | |
| --- | --- | --- | --- | --- |
| | process 1 (A↔B) | $k_{maj \to min}$ (s$^{-1}$) | process 2 (A↔C) | $k_{maj \to min}$ (s$^{-1}$) |
| S99T | $k_{ex,AB} = 188.3 \pm 9.5$ (s$^{-1}$), $p_B = 1.22 \pm 0.03$ (%) | 2.3 ± 0.1 | | |
| S99T/C115S | $k_{ex,AB} = 422.9 \pm 14.7$ (s$^{-1}$), $p_B = 1.93 \pm 0.05$ (%) | 8.2 ± 0.4 | $k_{ex,AC} = 177.7 \pm 16.1$ (s$^{-1}$), $p_C = 0.31 \pm 0.01$ (%) | 0.55 ± 0.05 |
| S99T/C115S/I97V | $k_{ex,AB} = 265.1 \pm 10.8$ (s$^{-1}$), $p_B = 4.32 \pm 0.09$ (%) | 11.5 ± 0.5 | $k_{ex,AC} = 186.5 \pm 18.1$ (s$^{-1}$), $p_C = 0.73 \pm 0.02$ (%) | 1.36 ± 0.1 |
| S99T + Suc-AFPF-pNA | $k_{ex,AB} = 118.3 \pm 37.0$ (s$^{-1}$), $p_B = 2.5 \pm 0.27$ (%) | 3.0 ± 1.0 | | |

**Fig. 2** Rescue mutants alter the conformational dynamics of CypA as measured by NMR. **a** Dynamics on WT CypA[7] (shown here) and S99T[8] (**c**) identified a slower (group-I, red) and faster dynamical process (group-II, blue). Residues Gly74 (blue) and Asn102 (red) are indicated with a sphere. **b** For the three mutants, $^{15}$N-CPMG dispersion profiles for a representative residue in fast exchange (Gly74, group-II) and slow exchange (Asn102, group-I)[8]. The fast-exchange process is virtually unaltered by the mutations, whereas $R_{ex}$ increases consecutively from single via double to triple-mutant (see also Supplementary Data 3). **c, e, g** Quantitative analysis of fast and slow protein dynamics of CypA rescue mutants by CPMG relaxation and CEST experiments are plotted onto the structure. Blue and red color coding of the cartoon representation denotes fast and slow dynamics, respectively, as determined from the temperature-dependence and shape of CPMG relaxation dispersion profiles (Supplementary Data 1 and 2). Spheres represent residues in slow exchange with quantifiable CEST profiles. **c** CEST data for all 15 residues (red spheres) in S99T can be globally fit to a two-site exchange process (Supplementary Fig. 3, Supplementary Data 4 and Table 2). **e, g** Residues with CEST profiles in S99T/C115S (**e**) and S99T/C115S/I97V (**g**) are well-described by two distinct slow processes (red and green, respectively), whereas residues shown in orange sense both processes and require a three-site exchange model (Supplementary Figs. 4 and 5, Supplementary Data 5 and 6 and Table 2). **d, f, h** Representative $^{15}$N-CEST profiles of residue Asn102, measured at the indicated field strengths are shown for single (**d**), double (**f**), and triple (**h**) mutants of CypA. The chemical shifts for the major (solid line) and minor states (– – – and – · – lines) are indicated. Uncertainties in $R_{2,eff}$ (**b**) are determined from the rmsd in the intensities of duplicate points ($n = 4$) according to the definition of pooled relative standard deviation; uncertainties in $I/I_0$ for CEST data (**d, f, h**) are determined from the rmsd in the baseline of the profile where no intensity dips are present (typically, $n > 50$)

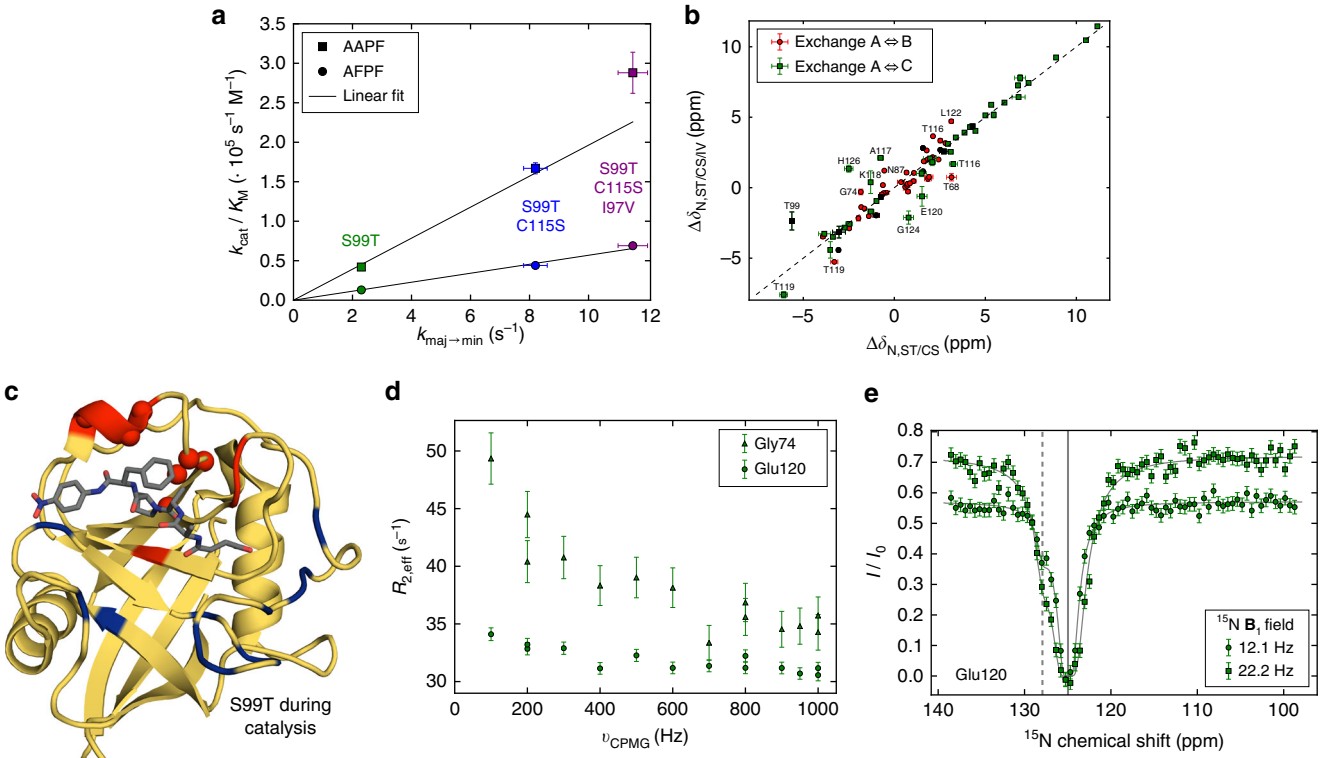

**Fig. 3** Protein dynamics during turnover and catalytic efficiency correlate. **a** Correlation between $k_{cat}/K_M$ and $k_{maj \to min}$ measured by CEST across all rescue mutations for apo protein fitted using orthogonal distance regression through the origin (AFPF: adj. $R^2 = 0.999$, $\chi^2_{red} = 1.1$; AAPF: adj. $R^2 = 0.995$, $\chi^2_{red} = 3.7$). **b** Correlation of the chemical shift differences between major and minor conformations for the two processes observed in S99T/C115S and S99T/C115S/I97V. Residues within 5 Å of mutation site (I97V) are shown in black and assignments are given if the variation in $\Delta\delta$ is >1.5 ppm. **c** Quantitative analysis of fast and slow protein dynamics of CypA S99T during catalysis of Suc-AFPF-pNA peptide (gray sticks) measured by NMR. CPMG relaxation dispersion experiments revealed, similarly as in apo S99T, fast motion mainly in the flexible loop (blue) and a slow process (red) consistent with the CEST data (spheres). **d** Representative CPMG profiles for a residue in fast (Gly74) and slow (Glu120) exchange and **e** CEST profile for Glu120 during catalysis (see Supplementary Data 7 and 8 for all profiles). Error bars in **a** and **b** denote the (propagated) standard errors in the fitted parameters. The uncertainties in CPMG (**d**) and CEST (**e**) data are determined as described in Fig. 2

64–74 loop[25]. The authors postulate that this loop dynamics is directly linked to catalysis via long-range concerted motions extending to the active site. This model is incompatible with the loop dynamics measured here, which remains fast for all CypA forms and is not correlated to catalytic turnover. In contrast, the group-I residues could be directly linked to catalysis via concurring changes in dynamics between WT, S99T, and the evolved enzyme forms, and catalytic turnover rates. This highlights the importance of NMR dynamics measurements and their quantitative comparison to corresponding changes in catalytic rates, which cannot be extracted from an ensemble-averaged NOE-based structure calculation. The slowness of conformational interconversion in CypA requires enhanced computational sampling methods for MD simulations. Transitions between the experimentally determined conformational states[8] were calculated using parallel-tempering metadynamics resulting in a free energy difference for S99T[33] that is in excellent agreement with the experimental value of −2.5 kcal/mol obtained here from the CEST data. A two-state ensemble for the *cis*- and *trans*-peptide bound CypA calculated by replica-exchange MD simulations in combination with NMR constraints[32] show only minimal protein conformational differences compared to the starting crystal structures (Supplementary Fig. 10). This is in sharp contrast to our NMR dynamics for WT[7] and S99T during catalysis measured here, which clearly shows that conformational sub-states interconvert across the core catalytic network and that this rate is correlated to catalysis.

It is notable that the function of CypA was modulated by mutations that do not directly contact the substrate. While the results of numerous "second shell" mutations emerging from directed evolution experiments have been interpreted based on speculative links between protein dynamics and changes in activity, experimental evidence of alterations of populations or kinetics of alternative conformations has been sparse[13,14]. Here, the increased dynamics can be rationalized by the ability of Phe113, which directly abuts the substrate, to transition between different rotameric states. This transition is controlled by the repacking of alternative conformations of core residues, such as Thr99, and is enabled by the decreased bulk of the mutated residues (Cys>Ser; Ile>Val). NMR spectroscopy directly reveals how the kinetics of these transitions between alternative internal packing arrangements is correlated with the increased catalytic activity accumulated during directed evolution. The slow dynamics associated with the catalytic cycle of CypA, and likely for many other enzymes, are only now becoming accessible to molecular simulations[34]. Therefore, the lessons of how non-active site mutations can alter conformational dynamics, derived from our integrated analysis of room-temperature X-ray data and NMR dynamics measurements, could be applied more broadly to aid the long-term goal of rationally improving enzymes generated by directed evolution.

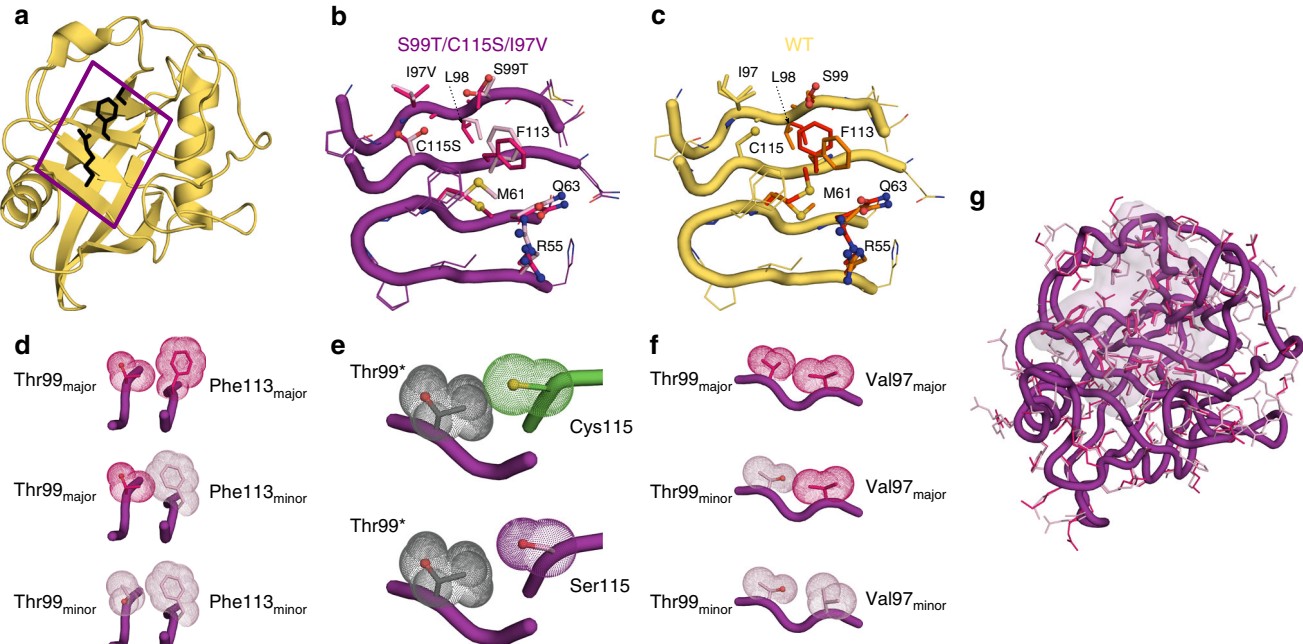

**Fig. 4** Structural basis of the increased protein dynamics from room-temperature X-ray crystallography on rescue mutants. **a** X-ray structure of CypA (2CPL[40]) with the active-site residues Arg55, Ser99, and Phe113 shown in black stick representation. The boxed area indicates the extended dynamic network shown in more detail in **b** and **c**. **b** Major and minor side chain conformations are shown for S99T/C115S/I97V (5WC7, 1.43 Å, see Supplementary Table 1) in purple and pink, respectively. The populations are flipped relative to wild-type CypA (3K0N, 1.4 Å)[8] (**c**), where major/minor states are shown in red and orange, respectively. The Phe113-in state corresponds to the major WT state and the minor S99T/C115S/I97V state. Correspondingly, the Phe113-out state corresponds to the minor WT state and the major S99T/C115S/I97V state. **d**, **e**, **f** Less steric hindrance due to the reduced size of side chains in rescue mutants facilitates the interconversion between major and minor conformations. Coupling between the conformation of Thr99 and Phe113 (**d**) and Val97 and Thr99 (**e**) are necessary to relieve clashes. The C115S (**f**, purple) mutation allows for a transition between Thr99 conformations (gray indicates morph between the major in minor state labeled Thr99*) without clash, in contrast to the bulkier Cys residue (green). **g** CONTACT analysis[28] of alternative conformations of S99T/C115S/I97V identifies a network extending across group-I residues (pink surface representation) consistent with the NMR results

## Methods

**Library screen**. Mutant libraries were created using error prone PCR as in Rockah-Shmuel et al.[35] For the initial library used to isolate S99T/C115S and subsequent screening efforts, the mutation rate was tuned to <3 new mutations per gene. Over 1000 individual clones were screened in the initial screen that identified S99T/C115S and an additional 1500 clones were screened in a second library to identify S99T/C115S/I97V. Subsequent screens of more than 5000 variants, including random mutations and focused libraries randomizing contacting residues, to identify additional mutations did not yield any new mutations with gains of function.

From transformations of >10,000 individual colonies, individual isolates were picked into 96-well blocks and grown overnight at 37 °C. For induction, 4 µl of the overnight culture was diluted into a fresh 96-well block containing 1 ml of LB and grown for 3 h prior to addition of 100 µM IPTG. The induced culture was then grown overnight at room-temperature and harvested by centrifugation at 3000×g for 15 min. The media was decanted from the 96-well block and 125 µl of lysis buffer (20 mM Tris, pH 8, 1% Triton S100) was added. The block was shaken for 30 min at room-temperature and then frozen at −80 °C.

Prior to assaying the AvrRpt2 activity, the 96-well block was thawed. To remove the membrane and unlysed cells, the thawed lysate was centrifuged at 3000×g for 15 min. Overall, 30 µl of the extract was transferred carefully to a 96-well plate. A master mix of inactive 0.25 mg/ml AvrRpt2 and 1 mM substrate peptide (Abz-IEAPAFGGWy-NH2, where y = 3-nitro-Tyrosine) were mixed in reaction buffer (20 mM HEPES, pH 8.5, 50 mM NaCl, 1 mM DTT). The reaction was initiated by mixing 10 µl of lysate with 30 µl of master mix in a Costar Black Flat bottom 96-well plate (Corning, product #3694) and measured in a Safire microplate reader monitoring fluorescence at 418 nm (excitation at 340 nm).

**Sample preparation**. CypA mutants were generated using site-directed mutagenesis on the CypA S99T plasmid using the Phusion High-Fidelity DNA Polymerase (New England Biolabs) and the primer sequences listed in Supplementary Table 3. Wild-type CypA and mutant proteins were essentially expressed and purified as described previously[6]. Briefly, LB medium or M9 minimal medium containing 2 g/l U-[13C]-D-glucose and/or 1 g/l 15NH4Cl (Cambridge Isotope

Laboratories, Tewksbury, MA, USA) as the sole carbon and nitrogen source were used to express unlabeled and 13C/15N-labeled or 15N-labeled CypA, respectively. Cells were grown at 37 °C to an $OD_{600}$ ~0.6 after which protein expression was induced with 0.3 mM IPTG for 4 h at the same temperature (wild-type and S99T) or overnight at 20 °C (S99T/C115S and S99T/C115S/I97V). Cells were lysed in 25 mM MES, pH 6.1, 5 mM β-mercaptoethanol and purified on a SP-Sepharose column using a NaCl gradient. Fractions containing CypA were pooled and dialyzed overnight into 50 mM $Na_2HPO_4$, pH 6.8, 5 mM β-mercaptoethanol. Remaining DNA and other impurities were removed using a Q-Sepharose column by collecting the flow-through. CypA was purified to homogeneity on a size-exclusion column (HiLoad 16/600, S75) equilibrated in 50 mM $Na_2HPO_4$, pH 6.5, 250 mM NaCl, 5 mM β-mercaptoethanol. Samples were dialyzed overnight into their final buffers (50 mM $Na_2HPO_4$, pH 6.5, 1 mM TCEP for NMR and 20 mM HEPES, pH 7.5, 100 mM NaCl, 0.5 mM TCEP for activity measurements, respectively). All purification steps were performed at 4 °C.

**Crystallography**. S99T/C115S/I97V and S99T/C115S CypA were produced and crystallized similarly to previous studies of wild-type CypA[8]. Briefly, crystals were grown by mixing equal volumes of well solution (100 mM (4-(2-hydroxyethyl)-1-piperazineethanesulfonic acid) HEPES, pH 7.5, 23% PEG 3350, 5 mM Tris (2-carboxymethyl) phosphine [TCEP]) and protein (60 mg/ml in 20 mM HEPES, pH 7.5, 100 mM NaCl, 0.5 mM TCEP) in the hanging-drop format. To collect the room-temperature synchrotron data set, paratone oil was applied to cover a 2 µl hanging drop containing a single large crystal. The crystal was harvested through the paratone and excess mother liquor was removed using a fine paper wick. Attenuated data were collected at ALS beamline 8.3.1 with a collection wavelength of 1.115 Å at 273 K controlled by the cryojet using an ADSC Q315r detector. Data were processed with XDS[36], monitoring scaling statistics to confirm a lack of radiation damage[37] and CC statistics for high-resolution cutoffs[38].

Molecular replacement was performed in Phaser[39] using 2CPL[40] as an initial search model. Residues were manually mutated in Coot[41] and subjected to multiple rounds of refinement using phenix.refine[42]. To add alternative conformations in a systematic manner, the refined single conformer model was rebuilt using qFit[27]. To finalize the model, further manual improvements to the connectivity of alternative

conformations and the ordered solvent molecules were performed iteratively with cycles of phenix.refine. Structure validation was performed using MolProbity and yielded the following statistics for S99T/C115S/I97V: Ramachandran (favored: 96%, allowed 4%, outliers: 0%), 1.4% rotamer outliers and a clashscore of 0.98 and S99T/C115S: Ramachandran (favored: 96.3%, allowed 3.7%, outliers: 0%), 1.8% rotamer outliers and a clashscore of 2.52.

Analysis of contacting residues using CONTACT was performed (parameters: $T_{stress}$ 0.35 and Relief 0.9), as for wild-type CypA[28]. Ensemble refinements were performed using phenix.ensemble_refinement[43]. All figures were prepared using PyMol[44].

**Enzyme activity.** $k_{cat}/K_M$ for the enzyme-catalyzed *cis*-to-*trans* isomerization of succinyl-AAPF-*p*-nitroanilide and succinyl-AFPF-*p*-nitroanilide (Suc-AXPF-pNA; Bachem) was measured at 10 °C using the standard chymotrypsin-coupled assay[24]. The increase in absorbance at 390 nm was fit to a single exponential to yield a rate constant for the interconversion between *cis*- and *trans*-peptide. Both the uncatalyzed, background (triplicates), and enzyme-catalyzed reaction were (average/ standard deviation of at least three different enzyme concentrations, each measured in triplicate) measured for both peptides and different enzyme concentrations chosen such that the rate of the enzyme-catalyzed reaction is between 3- and 15-fold faster than the uncatalyzed reaction.

**NMR spectroscopy and data analysis.** NMR experiments were recorded on an Agilent DD2 600 MHz four-channel spectrometer equipped with a triple-resonance cryogenically cooled probe-head or a Varian Unity Inova 500 MHz spectrometer equipped with a room-temperature triple-resonance probe. NMR samples contained 0.25 mM (for peptide $K_D$ experiments) or 1 mM (all other experiments) CypA in 50 mM $Na_2HPO_4$, pH 6.5, 1 mM TCEP, 0.02% $NaN_3$ and 10 (v/v) % $D_2O$. The CypA S99T + AFPF sample contained ~0.85 mM CypA and 10.5 mM Suc-AFPF-pNA. Sample temperatures were calibrated using the 4% methanol + 96% methanol-$d_4$ sample (DLM-5007, Cambridge Isotope Laboratories). All data sets were processed with the NMRPipe/NMRDraw software package[45] and visualized/ analyzed using the program NMRFAM-SPARKY[46].

**Backbone assignments.** TROSY-versions of a 3D HNCACB[47] and CBCA(CO) NH[48] experiments were recorded at 25 °C to obtain a nearly complete sequential backbone resonance assignment ($H^N$, N, $C^\alpha$, $C^\beta$) of CypA S99T/C115 and S99T/ C115S/I97V. The HNCACB experiments were acquired with 50 ($^{15}$N) × 70 ($^{13}$C) × 537 ($^1$H) complex points, with maximum evolution times equal to 22.2 ($^{15}$N) × 8.3 ($^{13}$C) × 64.0 ($^1$H) ms. An interscan delay of 1 s was used with 8 scans per transient, giving rise to a net acquisition time of 36 h. The CBCA(CO)NH experiments were acquired with 59 ($^{15}$N) × 50 ($^{13}$C) × 537 ($^1$H) complex points, with maximum evolution times equal to 26.2 ($^{15}$N) × 5.9 ($^{13}$C) × 64.0 ($^1$H) ms. An interscan delay of 1 s was used with 8 scans per transient, giving rise to a net acquisition time of 30 h. Cross peak assignments were propagated to CPMG and CEST experiments at lower temperatures using [$^1$H-$^{15}$N]-TROSY-HSQC[49,50] spectra recorded at different temperatures between 10 and 25 °C and where necessary confirmed using a 3D $^{15}$N-edited NOESY[51] data set recorded at 10 °C. Cross peaks for Val2 and Glu81 were not visible at 25 °C, but could be assigned from the data recorded at 10 °C. The chemical shift assignments of CypA S99T/C115S and S99T/C115S/I97V have been deposited in the BioMagResBank[52] with accession codes 27217 and 27218, respectively.

**$K_D$ measurements.** Dissociation constants for Suc-AFPF-pNA at 10 °C were obtained by titrating the peptide (final concentrations: 0, 1, 3, 5, 9 mM) into a solution of 0.25 mM CypA. Line-shape fitting was performed using PINT[53,54] to obtained cross peak positions in the individual spectra. The combined chemical shift difference $\Delta\delta$ was calculated according to Eq. (1):

$$\Delta\delta(ppm) = \left[ \Delta\delta_H^2 + (\Delta\delta_N/R_{scale})^2 \right]^{1/2},$$

where $R_{scale} = 6.3$ was determined according to Mulder et al.[55] Resonances with sufficient signal-to-noise and for which $\Delta\delta \geq 0.035$ ppm were included in the fits to determine dissociation constants using Eq. (2):

$$\Delta\delta = \Delta\delta_{sat-apo}$$
$$\frac{([AFPF] + [CypA] + K_D) - \sqrt{([AFPF] + [CypA] + K_D)^2 - 4 \cdot [AFPF] \cdot [CypA]}}{2 \cdot [CypA]},$$

where [CypA] is the total enzyme concentration, [AFPF] is the concentration of peptide and $\Delta\delta_{sat-apo}$ is the combined chemical shift difference between apo and saturated CypA. All resonances (13, 23, and 26 for CypA S99T, S99T/C115S, and S99T/C115S/ I97V, respectively) were fit simultaneously in Mathematica 11.2[56] and standard errors are obtained from the global fit. The solubility of the peptide and sample stability limits the highest achievable concentration in our titration experiments and we could only attain data up to 5 mM AFPF for the double- and triple-mutant.

We note that to calculate relative changes in $k_{cat}$, the assumption was made that the $K_D$ values obtained as described above are equal to the $K_M$.

**CPMG relaxation dispersion experiments.** Relaxation dispersion experiments on CypA were recorded on a 600 MHz spectrometer at different temperatures (10 and 15 °C for all mutants, and additionally at 20 °C for the triple-mutant). A TROSY-version of the relaxation-compensated $^{15}$N-CPMG pulse sequence[57,58] was used, with the CPMG period implemented in a constant-time manner[59].

The experiments were recorded as a series of 16 interleaved 2D data sets, with the constant-time relax period set to 40 ms. The CPMG field strengths were equal to 100, 200, 300, 400, 500, 600, 700, 800, 900, 950, and 1000 Hz, with duplicate experiments recorded for the reference experiment and 200, 800, and 1000 Hz. Experiments were acquired with 128–155 ($^{15}$N) × 536 ($^1$H) complex points, with maximum evolution times equal to 56.9–68.9 ($^{15}$N) × 63.9 ($^1$H) ms. An interscan delay of 2 s was used with 16, 24, or 32 scans per transient, giving rise to net acquisition times between 40–88 h for a complete pseudo-3D data set.

**CPMG data analysis.** Line-shape fitting was performed using PINT[53,54] and the obtained cross peak volumes were used to calculate the values of $R_{2,eff}$. Error estimation in the experimental data using the four duplicate data points was performed as described earlier by Mulder et al.[60] For ease of comparison, the values of $R_{2,inf}$ were calculated by taking the average of the $R_{2,eff}$ at the three highest $\nu_{CPMG}$ values (i.e., 900, 950, and 1000 Hz) and normalized to the lowest temperature. Data sets for each mutant at the different temperatures were fitted using PINT[53,54] to analytical equations for two-site exchange[61–63] and the *F*-test was used to determine whether a residue experiences exchange. If at any of the temperatures the exchange is determined to be significant ($p < 0.05$) for a residue, it is included in the respective supplementary data. To determine whether a residue is in the fast or slow-exchange regime, we compared the CPMG relaxation dispersion profiles at the different temperatures to estimate the exchange contribution, $R_{ex}$. The exchange contribution in slow-exchange depends solely on the rate constant for the major-to-minor transition (i.e., $R_{ex} \sim k_{maj\to min}$), whereas in the fast-exchange regime it is inversely proportional to the rate of interconversion (i.e., $R_{ex} \sim p_A\, p_B \Delta\omega^2/k_{ex})$[64]. Depending on the exchange regime, $R_{ex}$ will have different a temperature-dependence: $R_{ex}$ increases with temperature in the slow-exchange regime, whereas it decreases in the fast-exchange regime. In cases where the temperature-dependence was inconclusive, we determined the exchange regime from the overall shape of the dispersion profile: quenching of the exchange contribution already at low $\nu_{CPMG}$ values indicates a slow(er) process. Finally, if an exchange contribution was deemed significant in any of the data sets, this residue is included in the comparison between all mutant forms in Supplementary Data 3.

**CEST experiments.** $^{15}$N-CEST experiments[26] on CypA were recorded on a 500 MHz spectrometer at 10 °C for two (single- and triple-mutants) or three (double-mutant) different $^{15}$N $B_1$ field strengths. The weak irradiation fields applied during the relaxation delay were calibrated from 1D spectra as described by Guenneugues et al.[65] with the irradiation position set to an isolated cross peak that did not show exchange.

The experiments on CypA S99T were recorded with a $^1$H decoupling field strength of 2.3 kHz (using $90_x240_y90_x$ composite pulses) during the relaxation delay, $T_{EX}$, of 0.5 s. Two different $^{15}$N $B_1$ fields, 16.3 and 29.7 Hz, were used and a series of 2D data sets were acquired with $^{15}$N offsets ranging between 96.5 (97.7) and 140.4 (139.2) ppm, in 112 (85) increments of 20 (25) Hz for $\nu_1 = 16.3$ Hz ($\nu_1 = 29.7$ Hz), and one reference experiment. Each 2D data set comprised of 110 ($^{15}$N) × 512 ($^1$H) complex points, with maximum evolution times equal to 52.4 ($^{15}$N) × 64 ($^1$H) ms, an interscan delay of 1.5 s and eight scans per transient were used, giving rise to net acquisition times of about 118 h ($\nu_1 = 16.3$ Hz) and 94 h ($\nu_1 = 29.7$ Hz).

For CypA S99T in the presence of Suc-AFPF-pNA, two different $^{15}$N $B_1$ fields, 12.1 and 22.2 Hz, were used and a series of 2D data sets were acquired with $^{15}$N offsets ranging between 98.7 and 138.5 ppm, in 73 increments of 28 Hz, and one reference experiment. The experiments were recorded with a $^1$H decoupling field strength of 2.25 kHz (using $90_x240_y90_x$ composite pulses) during the relaxation delay, $T_{EX}$, of 0.4 s ($\nu_1 = 22.2$ Hz) or 0.5 s ($\nu_1 = 12.1$ Hz). Each 2D data set comprised of 110 ($^{15}$N) × 512 ($^1$H) complex points, with maximum evolution times equal to 52.4 ($^{15}$N) × 64 ($^1$H) ms, an interscan delay of 1.5 s and twelve scans per transient were used, giving rise to net acquisition times of about 112 h ($\nu_1 = 22.2$ Hz) and 116 h ($\nu_1 = 12.1$ Hz).

For CypA S99T/C115S, three different $^{15}$N $B_1$ fields, 12.2, 23.6, and 34.7 Hz, were used and a series of 2D data sets were acquired with $^{15}$N offsets ranging between 97.8 and 139.2 ppm, in 106 increments of 20 Hz, and one reference experiment. The experiments were recorded with a $^1$H decoupling field strength of 2.3–2.4 kHz (using $90_x240_y90_x$ composite pulses) during the relaxation delay, $T_{EX}$, of 0.4 s ($\nu_1 = 23.6$ Hz) or 0.5 s ($\nu_1 = 12.2$ and 23.6 Hz). Each 2D data set comprised of 110 ($^{15}$N) × 512 ($^1$H) complex points, with maximum evolution times equal to 52.4 ($^{15}$N) × 64 ($^1$H) ms, an interscan delay of 1.5 s and eight scans per transient were used, giving rise to net acquisition times of about 107 h ($\nu_1 = 23.6$ Hz) and 112 h ($\nu_1 = 12.2$ and 23.6 Hz).

The experiments on CypA S99T/C115S/I97V were recorded with a $^1$H decoupling field strength of 2.4 kHz (using $90_x240_y90_x$ composite pulses) during the relaxation delay, $T_{EX}$, of 0.4 s. Two different $^{15}$N $\mathbf{B_1}$ fields, 10.8 and 22.0 Hz, were used and a series of 2D data sets were acquired with $^{15}$N offsets ranging between 97.8 and 139.2 ppm, in 106 (85) increments of 20 (25) Hz for $\nu_1 = 10.8$ Hz ($\nu_1 = 22.0$ Hz), and one reference experiment. Each 2D data set comprised of 110 ($^{15}$N) × 512 ($^1$H) complex points, with maximum evolution times equal to 52.4 ($^{15}$N) × 64 ($^1$H) ms, an interscan delay of 1.5 s and eight scans per transient were used, giving rise to net acquisition times of about 86 h ($\nu_1 = 22.0$ Hz) and 107 h ($\nu_1 = 10.8$ Hz).

**CEST data analysis**. Line-shape fitting was performed using PINT[53,54] and the obtained cross peak volumes were used to calculate the ratio $I/I_0$. The $^{15}$N-CEST profiles (ratio versus irradiation position) were analyzed using the Python package ChemEx v0.6 (available from https://github.com/gbouvignies/chemex), which numerically propagates the Bloch–McConnell equations[66], as described earlier[26]. Uncertainties in the ratio $I/I_0$ were estimated from the apparent scatter in the baseline of the CEST profiles as implemented in the ChemEx package, whereas uncertainties in the fitting parameters (i.e., rate constants, populations and chemical shift differences) were determined from either the covariance matrix or 400–500 Monte-Carlo runs.

We analyzed all the CEST profiles assuming the absence of exchange, a two-site exchange model and, if required, a three-site exchange model (only for CypA S99T/C115S and S99T/C115S/I97V). In the "no-exchange" situation, the values of $k_{ex}$, $p_B$, $\Delta\omega$, and $\Delta R_2$ were fixed at 0 and only $R_1$, $R_2$, and $I_0$ were fit on a per-residue basis. Residues that showed elevated $R_2$ values and for which the $\chi^2_{red}$ was significantly above 1, are likely candidates to experience an exchange process (Supplementary Figs. 3b, 4f and 5f).

For the two-site exchange model, initially only residues that showed a clear second dip or asymmetry were included. A per-residue fit was performed where in addition to the parameters above also $k_{ex}$, $p_B$, and $\Delta\omega$ were allowed to float.

The clustering of $k_{ex}/p_B$ values for this initial subset indicated a single exchange process for CypA S99T (Supplementary Fig. 3a), and a global fit was performed after updating the initial $k_{ex}$, $p_B$, and residue-specific $\Delta\omega$ values. As a third step, we fixed the global exchange parameters, ($k_{ex}$ and $p_B$) and re-fitted all residues. Residues that had a fitted value of $|\Delta\omega| \geq 1$ ppm, an improved $\chi^2_{red}$ and consistent fitting parameters in Monte-Carlo runs (not fixing $k_{ex}$ and $p_B$), were included in the list of probes experiencing exchange. Finally, a global fit and 500 Monte-Carlo runs were performed with every exchanging residue and now allowing all parameters [$k_{ex}/p_B$ (global) and $\Delta\omega$, $R_1$, $R_2$, $\Delta R_2$, and $I_0$ (residue-specific)] to float. The results of this Monte-Carlo analysis are shown in Supplementary Fig. 3c for $k_{ex}/p_B$ and chemical shift differences are plotted on the structure (Supplementary Fig. 3d). Furthermore, assuming a two-state exchange model, the $\chi^2_{red}$ and fitted $R_2$ values for exchanging residues went down to their expected values, with the exception of the loop region undergoing exchange on the millisecond timescale that still exhibits high $R_2$ values (Supplementary Fig. 3b).

On the contrary, for CypA S99T/C115S and S99T/C115S/I97V the clustering of $k_{ex}/p_B$ values for the initial subset suggested the presence of two different processes (Supplementary Figs. 4a and 5a). Residues that seemingly fitted well to a two-site exchange model were grouped into two clusters and separately fitted to a global process, yielding two different rates of interconversion and populations (Supplementary Figs. 4b and 5b). In a similar manner as for CypA S99T we separately fixed the two combinations of global exchange parameters and re-fitted all data. Residues that had a fitted value of $|\Delta\omega| \geq 1$ ppm, an improved $\chi^2_{red}$ and consistent fitting parameters in Monte-Carlo runs (not fixing $k_{ex}$ and $p_B$), were included in the appropriate cluster of probes experiencing exchange. The two clusters were obtained consistently even when $k_{ex}/p_B$ values from the other one were used as starting points of the minimization. The residues present in the clusters are found in two different parts of the protein (cf. green and red spheres in Fig. 2e, g) and also have different $R_{ex}$ contributions in the CPMG relaxation dispersion profiles (Supplementary Data 1 and 2). Furthermore, CEST data for several residues, including F88 and N102, could not be fit to a two-site exchange model (Supplementary Figs. 4c and 5c).

Taken together, these observations show that the CEST data for the double- and triple-mutant require a three-state exchange model to explain the data. We note that for some residues (cf. orange spheres in Fig. 2e, g), the fast loop motion and slow processes give rise to convoluted exchange data. However, this does not automatically mean that any (or all) of these processes are correlated; it can simply be that a probe participating in one exchange process "feels" the fluctuating field of another, nearby exchange phenomena.

There are several possibilities to connect three different states (A = ground state, B = minor state 1, and C = minor state 2): one triangular model, where all states are connected, and three linear models (A↔B↔C, A↔C↔B, and B↔A↔C) as described earlier by Sekhar et al.[67] We did not consider the triangular model as it requires one additional fitting parameter and our NMR data is not of high enough quality to warrant this. From fitting the individual residues, we already obtained two values for $k_{ex}$/population (Supplementary Figs. 4b and 5b) and $\Delta\omega$ values, and those were used as starting values in the linear models. The combination with highest population was used as $k_{ex,AB}/p_B/\Delta\omega_{AB}$ and the other as $k_{ex,AC}/p_C/\Delta\omega_{AC}$.

The ACB-model, where the least populated minor state is the intermediate, did not fit the data; whereas the other two models gave very similar results and seem to describe the data equally well. On the basis of our current NMR and crystallography experiments and earlier data on CypA S99T[8], we favor the B↔A↔C model, where both processes are independent and this model has been used for further analysis.

For a number of CEST profiles, including F88 and N102, we do see two dips/asymmetry in the CEST traces (Supplementary Figs. 4d and 5d) and can, therefore, determine the chemical shift of all three states. The CEST data for these residues could be fitted to the BAC-model, with rate constants and populations very similar obtained for the "pure" two-state exchange models, indicating that the observed profiles are caused by these two, independent, slow-exchange processes. For others only one dip was visible, and there we determined the best initial value for $\Delta\omega$ of the second minor state by starting from the two possible options (i.e., the chemical shift of the other minor state is either close to the major state or to observable minor state) and comparing the $\chi^2_{red}$ value.

Finally, a global fit and 400–500 Monte-Carlo runs were performed with every exchanging residue and now allowing all parameters [$k_{ex,AB}/p_B$; $k_{ex,AC}/p_C$ (global) and $\Delta\omega_{AB}$, $\Delta\omega_{AC}$, $R_1$, $R_2$, $\Delta R_2$ and $I_0$ (residue-specific)] to float. The results of this Monte-Carlo analysis are shown in Supplementary Figs. 4e and 5e for $k_{ex}$/population and chemical shift differences are plotted on the CypA structure (Supplementary Figs. 4f and 5f). Furthermore, assuming a three-state exchange model, the $\chi^2_{red}$ and fitted $R_2$ values for exchanging residues went down to their expected values, with the exception of the loop region undergoing exchange on the millisecond timescale that still exhibits high $R_2$ values (Supplementary Figs. 4f and 5f).

**Data availability**. Structure factors and refined model of CypA S99T/C115S/I97V and S99T/C115S has been deposited in the PDB under accession codes 5WC7 and 6BTA, respectively. The NMR assignments of CypA S99T/C115S and S99T/C115S/I97V have been deposited in the BMRB under accession codes 27217 and 27218, respectively. Other data are available from the corresponding authors upon reasonable request.

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

## Acknowledgements

We thank N. Ollikainen, T. Kortemme, and H. van den Bedem for discussions. We are grateful to Drs. L.E. Kay, R. Muhandiram, and G. Bouvignies for sharing their NMR pulse sequence codes and help with the ChemEx software. F. Pontiggia is acknowledged for his help to implement the NMR data fitting workflow on the Brandeis HPC cluster. We thank G. Coaker for AvrRpt2 constructs. This work was supported by the Howard Hughes Medical Institute, the Office of Basic Energy Sciences, Catalysis Science Program, U.S. Dept. of Energy, award DE-FG02-05ER15699 and NIH (GM100966) to D.K., and a Searle Scholar Award from the Kinship Foundation, a Pew Scholar Award from the Pew Charitable Trusts, a Packard Fellowship from the David and Lucile Packard Foundation, NIH (GM110580), and NSF (STC-1231306) to J.S.F. R.O. was supported as an HHMI Fellow of the Damon Runyon Cancer Research Foundation (DRG-2114-12). Data collection at BL831 at the Advanced Light Sources is supported by the Director, Office of Science, Office of Basic Energy Sciences, of the U.S. Department of Energy under Contract No. DE-AC02-05CH11231, UC Office of the President, Multicampus Research

Programs and Initiatives grant MR-15-32859, and the Program Breakthrough Biomedical Research, which is partially funded by the Sandler Foundation.

## Author contributions

R.O., L.L., D.K., and J.S.F. designed experiments. L.L. and D.M. performed the library screen and developed the assay for screening activity in cell lysate with supervision of D. S.T. and J.S.F. L.L. performed the activity assay and R.O. analyzed the data. L.L., L.R.K., and J.S.F. performed the X-ray experiments; R.O. and M.W.C. performed the NMR experiments and R.O. analyzed the data. R.O., J.S.F., and D.K. wrote the paper. All authors contributed to data interpretation and commented on the manuscript.

## Additional information

**Competing interests:** The authors declare no competing interests.

