## [Peer Review File · Nature Communications]

Reviewers' comments:

Reviewer #1 (Remarks to the Author):

What are the major claims of the paper?

The major claim of this manuscript is that clustering residues showing increased dynamics are responsible for the observed increase in catalytic efficiency for improved CypA mutants selected from a directed evolution study using a catalytically impaired mutant as template.

Are they novel and will they be of interest to others in the community and the wider field?

This study would be of interest to the protein engineering and structural biology fields. It would, however, only be of interest to the wider community if the observed catalytic improvements of the mutants were significant. They are not. In addition, this work would be of interest to the wider community if the results were generally applicable to other enzyme and protein systems, which remains to be demonstrated. Although the study is interesting, the writing suffers from gross exaggerations and broad generalization that would need to be addressed and significantly adjusted prior to publication. While the results warrant publication, I am not entirely convinced that they should appear in Nature Communications.

If the conclusions are not original, it would be helpful if you could provide relevant references.

Provided that the authors readjust their message and address a number of important issues, these conclusions could be considered original. A long-standing goal in the protein engineering community is to demonstrate and control protein dynamics to affect/modulate/alter/improve protein function. To this day, not many reports have illustrated a direct evidence linking protein dynamics with improved enzyme catalysis, especially in directed evolution reports. The problem is I am not entirely convinced that this report demonstrates such a link. The authors certainly spin their message to try and convince the reader of the existence of such a link, but their raw results and analyses do not always support the overall claim and overly exaggerated enthusiasm.

Is the work convincing? On a more subjective note, do you feel that the paper will influence thinking in the field? Please feel free to raise any further questions and concerns about the paper.

Using a number of new and forward-looking methodologies (CPMG, CEST, RT crystallography), the authors make a nice characterization of major-to-minor state population shifts observed in solution for a selected few mutants obtained from directed evolution. It is, however, harder to be convinced that their dynamics and structural observations are directly responsible for the modest catalytic improvements in these enzyme variants. Their results remain correlated observations that do not seem to indicate causation. While demonstrating causation is certainly a difficult task, I am not convinced it has been achieved here.

Comment on the appropriateness and validity of any statistical analysis, as well the ability of a researcher to reproduce the work, given the level of detail provided.

I am generally concerned by the exaggerated enthusiasm in the writing of this manuscript, and conclusions not entirely supported by the results. While it is technically 'true' that the authors have selected mutations that 'rescue' part of the catalytic activity in CypA using directed evolution, careful examination of the raw kinetics parameters, NMR data, and analyzed results offers a much more modest portrait of reality.

1) The comparative analysis of catalytic efficiencies (k_{cat}/K_m) for WT and mutants for both peptide substrates illustrate how modest the directed evolution rounds actually selected for 'rescued' catalysis (i.e. improved variants over a catalytically impaired mutant) (Figure 1e). While WT CypA shows a $k_{cat}/K_m = 40 \times 10^5 \text{ M}^{-1}\text{s}^{-1}$ and the catalytically impaired S99T 'dead' mutant a $k_{cat}/K_m = 0.1 \times 10^5 \text{ M}^{-1}\text{s}^{-1}$ (400 times lower catalytic efficiency), both rescued mutants only show modest 5- to 7-fold improvements relative to S99T. Results with both peptide substrates yields similarly minor catalytic improvements for the 'rescued' mutants. Also, for some reason these comparisons are only presented in catalytic efficiency (k_{cat}/K_m). The authors never show individual k_{cat} and K_m parameters. This is puzzling and concerning.

2) Authors mention: "Extensive efforts to further improve enzymatic activity by many more rounds of evolution were unsuccessful." While they hypothesize that this could mean that they are stuck in a local minimum of the fitness landscape, this could also suggest that CypA variants have reached the limit of catalytic improvements in the context of the S99T mutation. Considering that substrates show very low substrate affinities (K_d) and that modest improvements in catalytic efficiency are observed in the rescued mutants, wouldn't this also unfortunately demonstrate that the contribution of residue dynamics on the overall catalytic cycle in CypA is minimal at best?

3) The authors mention that the two mutations function by modulating the turnover rate rather than substrate binding, presenting similar binding affinities (K_d) for all rescued mutants. I am concerned by the fact that all K_d values relate to extremely low mM binding affinities for all mutants, especially in light of the fact that the WT CypA K_d is not presented or even discussed anywhere in the manuscript, and presented in comparison with mutant values.

4) Showing a single "representative" dynamic residue for groups I and II in Figure 2b is hardly convincing for the rather bold statement the authors are trying to convey. Allegedly, the reader is supposed to refer to Figure 2a and Supplementary Data 3 to observe increased Rex values for group I residues and similar fast motion for all enzyme forms for group II residues (65-80). However, both the "representative" G74 and N102 residues of Figure 2b are not highlighted on the CypA structure in Figure 2a (we don't know where they are), nor are the red group I residues in any way labeled or listed in Supplementary Data 3. This latter supplementary file shows ^{15}N -CPMG relaxation dispersion for all residues of CypA and for all mutants. However, the reader does not know which residues were considered part of the red or blue residue groups (or most importantly how they were selected), and therefore cannot verify the authors' claims. Also, many relaxation dispersion curves show significant error bars, very low ΔR_2 values, or simply 'bad curves' data, suggesting that the authors cherry picked a limited number of residues to illustrate a point that is not necessarily generalized among grouped residues. The authors should illustrate several additional representative relaxation dispersion curves in Figure 2b, in addition to clearly explaining and listing which residues were selected to be part of which residue grouping in all Supplementary Data files. Although not as significant, this comment should also be considered for CEST residue selection.

5) Why are k_{cat} values never presented for all peptide substrates? These values should be presented and analyzed, both in relationship with WT CypA and comparatively between all mutants. The authors mention: "Substrate binding affinities are only slightly changed relative to the S99T mutant, suggesting that the two mutations function by modulating the turnover rate rather than substrate binding." Yet, we are only presented with catalytic efficiency (k_{cat}/K_m) comparisons and their linear correspondence with the slow interconversion between major and minor states. If the mutations truly affect catalysis, the direct contribution to k_{cat} should also be presented (and not just k_{cat}/K_m).

6) The authors are quick to point out a "remarkable" linear correspondence between the rate of

interconversion between the major and minor states and the k_{cat}/K_m parameter, further suggesting that this observation "corroborates our hypothesis that the increased dynamics of group-I residues is indeed responsible for the rescue in enzymatic activity." This is a significantly bold shortcut that is far from a demonstration of causality. No matter how linear and nice the correlation is, correlation does not imply causation. I am still not convinced of the causal link between dynamics and function in this case, especially considering that the authors never explain the precise atomic-scale molecular mechanism that directly links the changes in dynamics with the functional effects. This is quite surprising considering that they even have RT crystals to hypothesize on how these structural/dynamical changes could be linked to catalysis in the active site (i.e. substrate repositioning, discrimination, recognition, release, etc.). Perhaps performing additional mutations that would correspondingly translate into proportional changes in catalytic function and/or demonstrating the temperature dependence changes between dynamics and function would help.

7) Aiming to verify whether mutants populate the major and minor states in presence and absence of substrate like WT CypA, the authors point out that these "experiments proved to be difficult and only possible for S99T". Yet, based on the S99T results alone, they nevertheless conclude that: "Together these data show that the intrinsic dynamics in the group-I residues in the mutants are rate-limiting for the catalytic cycle." How can they generalize this behavior for all mutants if they have not performed the experiments for 2 out of the 3 mutants?! Additionally, they illustrate this behavior on the catalytically impaired S99T variant, which is not even one of the 2 'rescued' mutants they obtained in their directed evolution study. This is yet another significant logical shortcut and oversimplification/generalization in their argumentation.

Reviewer #2 (Remarks to the Author):

Overall I think this is a nice paper and makes an important contribution to the literature. The point of difference is that whereas other papers have had to infer changes in dynamics by analysis of crystal structures and changes in conformational sampling, by looking at this compact model system, the authors have been able to directly look at the motions in vitro.

The paper is very (probably too) concise (which I understand owing to space restrictions in some journals) but Nat Comms doesn't have such restrictions and I think the paper would substantially benefit from expansion to clarify some of the points. These are detailed below, additionally, to separate this from a something better suited to a specialist NMR journal, I wonder if there might be some simpler ways to illustrate some of the NMR results to make it more accessible to non-NMR experts (like myself).

Specific points (apologies if these suggestions are because I've missed or misread something):

The introduction is very brief and probably misses some key references. One obvious one (to me) was Campbell et al (Nat Chem Biol 2016), which has looked at how protein dynamics change across an evolutionary trajectory. On reading the introduction, one gets the sense that no studies of this type have been performed, i.e. in addition to papers where protein dynamics have been speculated to have contributed to functional improvements, some (such as Campbell et al) have looked at this in some detail.

The first section, which describes the selection and screening method (an impressive technical achievement) and the activities of the variants would benefit from additional explanation. Ln 48, it would be good to list the magnitude of reduction in turnover due to the S99T mutation (x-fold lower) to provide an idea of the change. Indeed, it's normal to include a kinetics table, where the rates can

be listed (with errors) - I see it is provided in Fig 1, but without a ruler and a magnifying glass it is hard to work out what the numbers are. It might be in the supp/extended data but I couldn't find it referenced in the main text, nor in the files to download. Ln56, the library screening is covered very briefly, I know it is in more detail in the methods but it would serve to reader to describe the size of the library, the mutation rate, etc., to get an idea of the directed evolution. Ln 64 - it's stated each mutation contributes additively, but is this tested - what do the mutations do in the WT or S99T background? It is certain there is no epistatic relationship? Ln 63, probably my biggest concern is that the changes are actually quite small and this is not alluded to in the main text, it is only apparent when looking at the data. It appears that the S99T mutation causes a ~300-fold reduction, then the mutations increase this ~2-3 fold per round, but the final variant is still ~40-fold less active than the WT? This should be covered more clearly, because it affects some of the claims later on (not necessarily negatively, but it would be easier to follow if this is clear in the next sections).

I can't judge the technical aspects of the NMR experiments as I'm not an expert in that area, but I can comment on the general conclusions. For me, the biggest issue is the claim of correlation - but the WT is omitted from most of this analysis for some reason. What would be helpful would be some comparison between WT and S99T to demonstrate and quantify how the dynamics have changed and how this correlates with the 300-fold loss of activity. For instance Ln 94-95 the interconversion rate increases from 2.3, to 8.2, to 11.5, which sort of correlates with the increase in k_{cat}/K_m from ~0.4 to ~1.5 to ~2.5. But what is the interconversion rate for WT? Is it 600? It might be in previous papers and was omitted for space issues, but it would really strengthen the conclusions a lot if it can be shown that WT is orders of magnitude faster, like the rate. This would remove any doubt there might be around looking at what are pretty small changes across the S99T>triple mutant trajectory. Ln 99, the correspondence probably isn't remarkable, there are no r^2 values given and it's only three points, it looks excellent for *afpf* (but the changes are very small) and ok for *aapf*. But again, why isn't WT included in this, does it also correlate? It might be that there are other factors (in addition to dynamics) that cause the loss of activity in S99T - this would be interesting and wouldn't detract from the overall argument. In fact, it might be surprising if 100% of the changes observed can be attributed solely to these slow dynamics (maybe some mention of this could be made in the discussion).

In the caption to Figure 4, there is no caption for d, and there is a caption for h, which isn't in the figure.

Ln 128, the Phe113 in conformation is introduced, but it isn't discussed what its relevance is? Is there a functional effect of it being in or out? This section could use a little bit more elaboration.

Ln 183 states experimental evidence of alterations of populations of alternative conformations have been sparse - Campbell et al (NCB 2017) looked at this.

So, overall, expansion of some explanation, a few more refs?, more detail about activities, and inclusion of WT in the correlation analysis/argument, discussion of whether all this change is 100% attributable to dynamics, and some easily digestible figure to quantify loss of dynamics from WT to S99T to the rescued mutant would potentially improve the paper for a general audience. That being said, I do think it is an important piece of work that is suitable for a journal such as this.

I'm happy for this review to be non-anonymous and for the authors to contact me if anything needs to be clarified, Colin Jackson.

Reviewer #3 (Remarks to the Author):

The manuscript by Otten and co-workers presents experimental and computational data that provide evidence for a link between enzyme evolution and protein dynamics. In particular, directed evolution was applied to generate mutants (S99T/C115S and S99T/C115S/I97V) of an impaired variant (S99T) of the human proline isomerase cyclophilin A (CypA) in which catalytic activity is partially restored. NMR and room-temperature (RT) X-ray crystallography were then applied to examine side-chain dynamics and conformational heterogeneity, respectively. The increase in the catalytic rate of the S99T/C115S/I97V mutant is proposed to be due to accelerated interconversion between two functionally essential substates.

The thorough study has been carried out by pioneers and world-experts in the fields of directed enzyme evolution, NMR and RT protein crystallography and data analysis and is a beautiful example of synergistically using NMR and crystallography to elucidate the link between macromolecular structure, dynamics and function. It will be of great interest to the broad readership of Nature Communications. The manuscript would benefit from X-ray crystallographic analysis of S99T/C115S and S99T/I97V mutants (see below) but can be published without these additional data.

Detailed remarks in order of appearance in the text:

Line 64 : ' ... increase in activity ...' it might be adequate mentioning that the catalytic activity of S99T/C115S/I97V is only 3% of that of the WT, thus quantifying the 'partial restoration' mentioned in the abstract.

Line 72: '... suggested a direct link between the speed of a conformational change in a dynamic network ...' and what ? Please clarify.

Lines 126-128 : 'Both the C115S and I97V mutations subtly reduce the amino acid size, and combined partially restore the "Phe113-in" conformation as a minor conformation that can now be directly observed in the electron density (Fig. 4b,d-f).'

Without crystallographic analysis of S99T/C115S and S99T/I97V mutants it does not seem justified to state that ' ... both mutations ... combined partially restore the "Phe113-in" conformation as a minor conformation ...'. How can the authors exclude that one of the two mutations is sufficient to occupy the minor Phe113 conformation at a 20% level as is the case in the S99T/C115S/I97V mutant ? If the authors' hypothesis is true, Phe113 should be occupied at less (half ?) than 20% occupancy in both S99T/C115S and S99T/I97V mutants. Maybe such low occupancies would be difficult to determine accurately ? If the authors cannot include X-ray analysis of at least one of the two mutants, they should at least nuance their statement in lines 126 - 128.

Line 128: ' ... Phe113-in conformation ...' has not been introduced before and its identity and importance only becomes clear when studying fig. 4 in detail. It would already help to name and identify this conformation in the legend of fig. 4.

Line 128: ' ... minor conformation ...' This statement is only quantified (20% occupancy) in the Methods section. Given the suggested catalytic importance of the minor conformation of Phe113, it might be worth to consider mentioning occupancies of the WT, the S99T and the S99T/C115S/I97V mutants in the main text.

Line 145: 'Why did additional rounds of directed evolution fail to yield further improvement in the catalytic rate?' Even if DE failed, did the authors consider rational approaches ? In particular, the

referee wonders if avoiding the steric clash between Thr99minor and Val97major (Fig. 4f) in a S99T/C115S/I97A mutant would increase the catalytic rate beyond the one of S99T/C115S/I97V ?

Line 193: What is meant by `... long-term goal of rationally improving directed evolution.` ? How can `directed evolution` be `rationally improved` ? Better: `rationally improving enzymes generated by directed evolution.` ?

Line 247: lacks `respectively`

Legend fig 4: panel (d) is not described. Panel (h) should be panel (g). What is meant by `major in minor state` ?

Extended Data Figure 8: is the gain in conformational heterogeneity of Phe113 in S99T/C115S/I97V wrt S99T significant ? If not, please consider removing the time-averaged ensemble refinement.

We have addressed the thoughtful comments by all three reviewers in a revised manuscript. In general, the major points are a change of language (reviewer 1), an expanded discussion of epistasis (reviewer 2), the inclusion of the S99T/C115S mutant (reviewer 3), and a speculative assessment of how the changes relate to the catalytic mechanism (all reviewers). In this letter, we provide a point-by-point answer to all comments. Our comments are offset from the reviewer comments in green, with highlights of added text to the manuscript in red. We followed your guidelines for the uploaded revised manuscript, in which all changes to the original manuscript are highlighted in red.

Reviewers' comments:

Reviewer #1 (Remarks to the Author):

What are the major claims of the paper?

The major claim of this manuscript is that clustering residues showing increased dynamics are responsible for the observed increase in catalytic efficiency for improved CypA mutants selected from a directed evolution study using a catalytically impaired mutant as template.

Are they novel and will they be of interest to others in the community and the wider field?

This study would be of interest to the protein engineering and structural biology fields. It would, however, only be of interest to the wider community if the observed catalytic improvements of the mutants were significant. They are not. In addition, this work would be of interest to the wider community if the results were generally applicable to other enzyme and protein systems, which remains to be demonstrated. Although the study is interesting, the writing suffers from gross exaggerations and broad generalization that would need to be addressed and significantly adjusted prior to publication. While the results warrant publication, I am not entirely convinced that they should appear in Nature Communications.

We have significantly altered the wording of the manuscript and hope that the reviewer finds the more conservative and nuanced interpretations in the revised version acceptable (see especially the next point).

If the conclusions are not original, it would be helpful if you could provide relevant references.

Provided that the authors readjust their message and address a number of important issues, these conclusions could be considered original. A long-standing goal in the protein engineering community is to demonstrate and control protein dynamics to affect/modulate/alter/improve protein function. To this day, not many reports have illustrated a direct evidence linking protein dynamics with improved enzyme catalysis, especially in directed evolution reports. The problem is I am not entirely convinced that this report demonstrates such a link. The authors certainly spin their message to try and convince the reader of the existence of such a link, but their raw results and analyses do not always support the overall claim and overly exaggerated enthusiasm.

As stated above, we have reworded the conclusions to be more nuanced and conservative. As the reviewer states, there are no convincing reports, despite much speculation, of a causal link of protein dynamics and improved catalysis – and our revision places our work as an important piece of correlative evidence, while acknowledging that there is still much work to be done. For example, in the abstract, we replace the phrase “could be rationalized entirely” with “is correlated with”:

The increased catalysis selected for in the evolutionary screen is correlated with an accelerated interconversion between the two catalytically essential conformational sub-states, which are both captured in the high-resolution X-ray ensembles at room temperature.

And in the discussion, we replace “controls” with “is correlated with”:

NMR spectroscopy directly reveals how the kinetics of these transitions between alternative internal packing arrangements is correlated with the increased catalytic activity accumulated during directed evolution.

Is the work convincing? On a more subjective note, do you feel that the paper will influence thinking in the field? Please feel free to raise any further questions and concerns about the paper.

Using a number of new and forward-looking methodologies (CPMG, CEST, RT crystallography), the authors make a nice characterization of major-to-minor state population shifts observed in solution for a selected few mutants obtained from directed evolution. It is, however, harder to be convinced that their dynamics and structural observations are directly responsible for the modest catalytic improvements in these enzyme variants. Their results remain correlated observations that do not seem to indicate causation. While demonstrating causation is certainly a difficult task, I am not convinced it has been achieved here.

As highlighted above and in other minor changes throughout the manuscript, the revision has a more nuanced and conservative interpretation about the correlative nature of the increases.

Comment on the appropriateness and validity of any statistical analysis, as well the ability of a researcher to reproduce the work, given the level of detail provided.

I am generally concerned by the exaggerated enthusiasm in the writing of this manuscript, and conclusions not entirely supported by the results. While it is technically 'true' that the authors have selected mutations that 'rescue' part of the catalytic activity in CypA using directed evolution, careful examination of the raw kinetics parameters, NMR data, and analyzed results offers a much more modest portrait of reality.

We hope the reviewer finds the revised writing satisfactory, that describes a trajectory of mutations isolated to rescue the function of S99T that have a correlated increase in conformational dynamics and catalysis.

1) The comparative analysis of catalytic efficiencies (k_{cat}/K_M) for WT and mutants for both peptide substrates illustrate how modest the directed evolution rounds actually selected for 'rescued' catalysis (i.e. improved variants over a catalytically impaired mutant) (Figure 1e). While WT CypA shows a $k_{cat}/K_M = 40 \times 10^5 \text{ M}^{-1}\text{s}^{-1}$ and the catalytically impaired S99T 'dead' mutant a $k_{cat}/K_M = 0.1 \times 10^5 \text{ M}^{-1}\text{s}^{-1}$ (400 times lower catalytic efficiency), both rescued mutants only show modest 5- to 7-fold improvements relative to S99T. Results with both peptide substrates yields similarly minor catalytic improvements for the 'rescued' mutants. Also, for some reason these comparisons are only presented in catalytic efficiency (k_{cat}/K_M). The authors never show individual k_{cat} and K_M parameters. This is puzzling and concerning.

The reviewer is correct in the observation that our rescue mutants only partially restore the catalytic activity of wild-type CypA, as we have also noted in the abstract and the main text. We agree that separating k_{cat} and K_M parameters would be very useful; unfortunately, this is not possible due limitations of the chymotrypsin-coupled assay that is the *de facto* assay to measure proline isomerases¹. The weak affinity towards the assay substrates together with the high extinction coefficient of pNA, prohibit measurements at substrate concentration above k_{cat}/K_M conditions. High pNA concentrations saturate the detector of the spectrophotometer as soon as chymotrypsin is added as it instantaneously cleaves the ~90% *trans*-peptide and obscures the detection of the small fraction of *cis*-peptide that is isomerized by cyclophilin A. We attempted to develop a HPLC-based assay to avoid such limitations, but did not succeed as there were other experimental limitations (e.g., the strong acidic conditions needed to quench the reaction did also cause the substrate to decompose). Therefore, we turned to NMR titration experiments to measure changes in binding affinities.

We added the text:

“The intrinsic limitation of this enzymatic assay is the inability to measure at substrate-saturating concentration and, therefore, k_{cat} and K_M cannot be separated. In an effort to determine whether the observed changes in k_{cat}/K_M are due to changes in turnover number or substrate affinity, substrate binding was detected directly by NMR titration experiments.”

2) Authors mention: “Extensive efforts to further improve enzymatic activity by many more rounds of evolution were unsuccessful.” While they hypothesize that this could mean that they are stuck in a local minimum of the fitness landscape, this could also suggest that CypA variants have reached the limit of catalytic improvements in the context of the S99T mutation.

Yes, there is a good possibility that the limit of catalytic improvements in the context of the S99T mutation has been reached (since this mutation was originally designed to make a dynamically crippled enzyme²). It was not our intention to exclude this possibility and we have added text in the revision to reflect this: “Alternatively, any further catalytic improvement via additional mutations may be precluded in the context of the severe S99T mutation.”

Considering that substrates show very low substrate affinities (K_d) and that modest improvements in catalytic efficiency are observed in the rescued mutants, wouldn't this also unfortunately demonstrate that the contribution of residue dynamics on the overall catalytic cycle in CypA is minimal at best?

The contribution of the interconversion process between the major and minor state quantitatively measured here for CypA variants, and for wild-type in a previous publication³ provides strong evidence that this interconversion is limiting the overall turnover for all enzyme forms:

1. When comparing wild-type CypA and S99T, protein dynamics changes from fast to slow on the NMR time-scale and is paralleled by a large decrease in k_{cat} between these two CypA forms².
 2. The increase in protein dynamics measured by NMR from S99T to S99TC115S to S99TC115S/I97V is fully correlated with the corresponding changes in catalytic turnover. Yes, the overall effect is moderate as described in the manuscript, but the measured differences are significant and well outside the experimental errors.

3) The authors mention that the two mutations function by modulating the turnover rate rather than substrate binding, presenting similar binding affinities (K_d) for all rescued mutants. I am concerned by the fact that all K_d values relate to extremely low mM binding affinities for all mutants, especially in light of the fact that the WT CypA K_d is not presented or even discussed anywhere in the manuscript, and presented in comparison with mutant values.

Thank you for this suggestion. Low substrate affinities are seen across the board for both wild-type and CypA mutants. Although the wild-type affinities had been published before, we realize that it is easier for the reader of this manuscript to have all numbers in one table here. We added Table 1 to provide these values.

Table 1 | Kinetic parameters and substrate affinity for wild-type CypA and mutants measured at 10 °C.

CypA variant	$k_{cat}/K_M (\cdot 10^5 \text{ s}^{-1}\text{M}^{-1})$		K_D (mM)	
	Suc-AFPF-pNA	Suc-AAPF-pNA	Suc-AFPF-pNA	Suc-AAPF-pNA
wild-type	42.82 ± 2.38	123.22 ± 9.95		1.8 ± 0.14 ^a
S99T	0.13 ± 0.01	0.42 ± 0.01	5.4 ± 0.5	6.7 ± 0.8 ^a
S99T/C115S	0.44 ± 0.01	1.67 ± 0.07	4.8 ± 0.4	
S99T/C115S/I97V	0.69 ± 0.02	2.88 ± 0.26	3.3 ± 0.2	
C115S		53 ± 8		
I97V		122 ± 12		
I97V/C115S		88 ± 9		

^a measured for Suc-AAPF-pNA at 6 °C as described earlier².

4) Showing a single “representative” dynamic residue for groups I and II in Figure 2b is hardly convincing for the rather bold statement the authors are trying to convey. Allegedly, the reader is supposed to refer to Figure 2a and Supplementary Data 3 to observe increased Rex values for group I residues and similar fast motion for all enzyme forms for group II residues (65-80).

However, both the “representative” G74 and N102 residues of Figure 2b are not highlighted on the CypA structure in Figure 2a (we don't know where they are), nor are the red group I residues in any way labeled or listed in Supplementary Data 3. This latter supplementary file shows 15N-CPMG relaxation dispersion for all residues of CypA and for all mutants. However, the reader does not know which residues were considered part of the red or blue residue groups (or most importantly how they were selected), and therefore cannot verify the authors' claims. Also, many relaxation dispersion curves show significant error

bars, very low delta R2 values, or simply 'bad curves' data, suggesting that the authors cherry picked a limited number of residues to illustrate a point that is not necessarily generalized among grouped residues. The authors should illustrate several additional representative relaxation dispersion curves in Figure 2b, in addition to clearly explaining and listing which residues were selected to be part of which residue grouping in all Supplementary Data files.

We acknowledge the reviewer's point that while all the data was available in the original submission as extended/supplemental data, we could have made it easier for the reader to comprehend. In an effort to do so, we have followed the suggestions and made the changes given below:

1. indicated residues Gly74 and Asn102 with spheres on the CypA structure in Figure 2a.
2. indicated whether a residue belongs to "group-I" or "group-II" in Supplementary DataSets 1-3 and 7 by adding this label to the figure title and color-coding it (red and blue, respectively) as is done in Figure 2.
3. added a description on the selection of residues, which was succinctly described in the legend of Figure 2, in the Materials and Methods section under "CPMG data analysis".
4. added a sentence to the main text that the CPMG profiles are only used for qualitative results, and that CEST profiles were necessary to obtain accurate rate constants and populations:

"However, rate constants and populations are often not accurately determined by CPMG profiles for processes in the slow exchange regime. We note that the fast loop motion of group-II (residues 65-80) is observed in all enzyme forms and remains essentially unaltered (Fig. 2a-c,e,g and Supplementary Data 1-3). For a quantitative understanding of the mechanism underlying the increased catalysis along the directed evolution trajectory, we applied a powerful NMR method for studying systems in slow exchange, chemical exchange saturation transfer (CEST) spectroscopy⁴."

Finally, we would like to answer the last part of this question about Supplementary DataSet 3, where CPMG profiles for all mutants are shown. We did not cherry-pick residues, but rather show here all residues that experience exchange in at least one of the CypA mutants at any of the temperatures. Sometimes the uncertainty in $R_{2,eff}$ is indeed high and in part, this is due to the fact that data was recorded at 10 °C. We have now added the data for all mutants at 15 °C for a side-by-side comparison as the data quality at that temperature is often better and the exchange contribution is larger in the case of slow exchange. Additionally, we want to clarify that for a number of residues quantifiable exchange is only observed for CypA S99T/C115S/I97V and not for the single- and double-mutants. For comparison/completeness we however still show these flat dispersions profiles. There are several residues in group-I and group-II exhibit identical trends as the ones shown in Figure 2B, and with the improved labeling of the panels these should now be easy to spot in Supplementary DataSet 3.

Although not as significant, this comment should also be considered for CEST residue selection.

We color-coded the figure titles of the residues in Supplementary DataSets 5 and 6 in the same way as the spheres in Figure 2e,g in the revised manuscript. In contrast to the CPMG data, the CEST data were globally fit to obtain accurate values for populations and rate constants and the analysis is described in the Materials and Methods section. Additionally, we note that even though we initially separate the residues into clusters to obtain populations and interconversion rates, data were fit globally to a three-site exchange model in the final stage.

5) Why are kcat values never presented for all peptide substrates? These values should be presented and analyzed, both in relationship with WT CypA and comparatively between all mutants. The authors mention: "Substrate binding affinities are only slightly changed relative to the S99T mutant, suggesting that the two mutations function by modulating the turnover rate rather than substrate binding." Yet, we are only presented with catalytic efficiency (kcat/Km) comparisons and their linear correspondence with the slow interconversion between major and minor states. If the mutations truly affect catalysis, the direct contribution to kcat should also be presented (and not just kcat/Km).

We have clarified these points in our answers to 1 – 3 (i.e., added a table and new paragraph in the main text).

6) The authors are quick to point out a “remarkable” linear correspondence between the rate of interconversion between the major and minor states and the k_{cat}/K_m parameter, further suggesting that this observation “corroborates our hypothesis that the increased dynamics of group-I residues is indeed responsible for the rescue in enzymatic activity.” This is a significantly bold shortcut that is far from a demonstration of causality. No matter how linear and nice the correlation is, correlation does not imply causation. I am still not convinced of the causal link between dynamics and function in this case, especially considering that the authors never explain the precise atomic-scale molecular mechanism that directly links the changes in dynamics with the functional effects. This is quite surprising considering that they even have RT crystals to hypothesize on how these structural/dynamical changes could be linked to catalysis in the active site (i.e. substrate repositioning, discrimination, recognition, release, etc.). Perhaps performing additional mutations that would correspondingly translate into proportional changes in catalytic function and/or demonstrating the temperature dependence changes between dynamics and function would help.

Please see our answers above for the same first part of the questions describing our changes to the manuscript to describe better the quantitative changes in the dynamics of group-I residues from wild-type to S99T to the double and triple mutant, paralleled by the same amount of changes in catalytic turnover. In regard to the second part about a structural hypothesis, we have added a sentence and Extended Data Figure 9 to the manuscript about the potential for distinct contacts of the two substrate forms with the two conformations of Phe113:

“... rewired the internal packing to increase the dynamics of Phe113, which likely makes distinct contacts with the *cis*- and *trans*- substrate (Extended Data Fig. 9), correlated with surrounding group-I residues during the catalytic cycle.”

Extended Data Figure 9 | The triple mutant restores the alternative conformation of Phe113, which can make distinct complementary interactions with the *trans*-substrate.

(a) The tight surface complementarity of CypA surface (shown in mesh) for the Phe-out conformation (orange) overlaid with the *trans*-substrate model from an NMR ensemble. (b) In contrast, the Phe-in state (red) leaves a void and has poorer surface complementarity to the substrate. These modeled results suggest that the transition between different conformations of Phe113 may stabilize distinct substrate states.

However, given that we have not captured this interaction directly experimentally (it is based on modeling the substrate ensemble determined by Vendruscolo *et al.*⁵ onto our room-temperature X-ray data), it is still premature to speculate further.

7) Aiming to verify whether mutants populate the major and minor states in presence and absence of substrate like WT CypA, the authors point out that these “experiments proved to be difficult and only possible for S99T”. Yet, based on the S99T results alone, they nevertheless conclude that: “Together these data show that the intrinsic dynamics in the group-I residues in the mutants are rate-limiting for the catalytic cycle.” How can they generalize this behavior for all mutants if they have not performed the experiments for 2 out of the 3 mutants?! Additionally, they illustrate this behavior on the catalytically impaired S99T variant, which is not even one of the 2 ‘rescued’ mutants they obtained in their directed evolution study. This is yet another significant logical shortcut and oversimplification/generalization in their argumentation.

We realize that the link between our measured NMR data, x-ray structures and kinetic data to our conclusions was not explained well enough to the reader. We added text in the revised manuscript to rationalize this link, including referencing to earlier published data that are relevant for the argumentation. We agree that it is much better to have everything discussed in this manuscript so that the reader does not have to read earlier papers for a full logic understanding of the conclusions here. Thank you for suggesting to strengthen the interpretation/discussion of our results. The major change includes this additional text:

“In wild-type CypA, overall turnover is dictated by the interconversion between the major and minor conformations as the dynamics measured on the substrate-bound enzyme coincides with the rate of *cis* to *trans* isomerization of the bound substrate^{3,6}. This suggests that the two distinct protein conformations are needed to accommodate the *trans*

and *cis* form of the substrate. Secondly, the conformational exchange occurs on similar timescales in both the apo enzyme and during turnover³. To confirm that such a correspondence between the measured dynamics in the apo and turnover protein holds true for the mutants, CPMG and CEST experiments were performed on S99T during enzymatic turnover of a substrate peptide. These experiments on the mutants proved to be difficult and only possible for S99T due to stability, and the weak affinity for the peptides allowed for a maximum of ~70% saturation. For S99T during catalysis, we indeed observe both fast loop movement, and slow conformational dynamics in the group-I residues, very similar to the apo protein (Fig. 3c-e and Table 2). We conjecture that the increase in $k_{\text{maj} \rightarrow \text{min}}$ in the rescue mutants is responsible for the increase in k_{cat}/K_M . Note that interconversion in the apo state (“recycling” of the enzyme from the *trans*- to the *cis*-binding conformation in the chymotrypsin assay) and/or in the substrate-bound form can be rate limiting. In summary, these data show that the intrinsic dynamics in the group-I residues are rate-limiting for the catalytic cycle, as the severe reduction of the protein’s interconversion rate parallels the reduction in k_{cat}/K_M (see Tables 1 and 2)^{2,3}.”

We think that additional mutations are not necessary to make a strong link between dynamics and catalytic turnover rates. We have shown it now for four different forms of CypA with full parallel changes: wild-type to S99T to S99T/C115S to S99T/C115S/I97V. The two distinct conformational forms and their relevance for accommodating the *trans*- and *cis*-forms of the substrate are now explained in the above response and new Extended Data Figure 9.

We want to reiterate that in most cases reported, enzymatic turnover is *slowed down* by mutations whereas *protein dynamics increased*, or protein dynamics was quenched yet the protein was still functional. Here, we characterize an *increase in enzymatic turnover*, which is a more difficult task, and this increase *coincides with an increase in a very specific interconversion between two protein conformations*.

We want to thank the referee for excellent suggestions to revise the manuscript with changes in writing and additional information added to the manuscript (see new figures and tables) to fully convince him/her about our results. We value this scientific dialog.

Reviewer #2 (Remarks to the Author):

Overall I think this is a nice paper and makes an important contribution to the literature. The point of difference is that whereas other papers have had to infer changes in dynamics by analysis of crystal structures and changes in conformational sampling, by looking at this compact model system, the authors have been able to directly look at the motions in vitro.

The paper is very (probably too) concise (which I understand owing to space restrictions in some journals) but Nat Comms doesn't have such restrictions and I think the paper would substantially benefit from expansion to clarify some of the points. These are detailed below, additionally, to separate this from a something better suited to a specialist NMR journal, I wonder if there might be some simpler ways to illustrate some of the NMR results to make it more accessible to non-NMR experts (like myself).

We thank the referee Dr. Colin Jackson for placing the impact of this manuscript into the field and excellent to the point suggestions for improving the writing of the manuscript for a broad readership. We value his feedback as an expert in protein evolution, acknowledging the important contribution of this work to the field of protein function, evolution and catalysis.

Specific points (apologies if these suggestions are because I've missed or misread something):

The introduction is very brief and probably misses some key references. One obvious one (to me) was Campbell et al (Nat Chem Biol 2016), which has looked at how protein dynamics change across an evolutionary trajectory. On reading the introduction, one gets the sense that no studies of this type have been performed, i.e. in addition to papers where protein dynamics have been speculated to have contributed to functional improvements, some (such as Campbell et al) have looked at this in some detail.

Yes, this short list of references came indeed from a number restriction. We are happy that we can now add additional nice publications to our introduction. The extended introduction provides a much better recognition of what has been shown in the field before our work, and new aspects this manuscript adds.

The first section, which describes the selection and screening method (an impressive technical achievement) and the activities of the variants would benefit from additional explanation.

Ln 48, it would be good to list the magnitude of reduction in turnover due to the S99T mutation (x-fold lower) to provide an idea of the change. Indeed, it's normal to include a kinetics table, where the rates can be listed (with errors) - I see it is provided in Fig 1, but without a ruler and a magnifying glass it is hard to work out what the numbers are. It might be in the supp/extended data but I couldn't find it referenced in the main text, nor in the files to download.

We thank the reviewer for the suggestion and have added a short paragraph on the comparison of wild-type CypA and the S99T mutant [see also reviewer 1; points 2 and 3]. In addition, two additional tables were included showing the values and uncertainties for k_{cat}/K_M and K_D (Table 1) and the rate constants and populations from the CEST experiments (Table 2).

Ln56, the library screening is covered very briefly, I know it is in more detail in the methods but it would serve to reader to describe the size of the library, the mutation rate, etc., to get an idea of the directed evolution.

Thank you for this suggestion. We have added these details into this section:

“We expressed a library of CypA S99T variants created by random mutagenesis tuned to 1-3 mutations per gene, added inactive AvrRpt2 to cell lysate and monitored the cleavage of an AvrRpt2 substrate⁷ (Fig. 1a). Besides revertants to wild-type CypA (Ser99), our initial screen of ~1000 variants identified a variant (S99T/C115S) with increased activity (Fig. 1b). A second round of ~1500 variants in the background of S99T/C115S CypA identified an additional mutation (I97V) with a further increase in activity. Extensive efforts to further improve enzymatic activity by many more rounds of evolution were unsuccessful. Both gain-of-function mutations are in proximity of Thr99, but not in direct contact with the peptide substrate (Fig. 1c). “

Ln 64 - it's stated each mutation contributes additively, but is this tested - what do the mutations do in the WT or S99T background? It is certain there is no epistatic relationship?

We assayed these mutants against Suc-AAPF-pNA and found the following relationship:

- C115S: $(53 \pm 8) \times 10^5 \text{ s}^{-1}\text{M}^{-1}$ – down ~2-fold from wild-type
- I97V: $(122 \pm 12) \times 10^5 \text{ s}^{-1}\text{M}^{-1}$ – within error of wild-type
- I97V/C115S: $(88 \pm 9) \times 10^5 \text{ s}^{-1}\text{M}^{-1}$ – down 25% from wild-type

We have added this information into Table 1 and additionally added this sentence in the discussion:

There is a weak epistatic relationship for C115S in particular: it decreases activity in the wild-type background and increases activity in the S99T background (Table 1).

However, to speculate much more heavily on this finding leads into the reviewers next point about the magnitude of the changes.

Ln 63, probably my biggest concern is that the changes are actually quite small and this is not alluded to in the main text, it is only apparent when looking at the data. It appears that the S99T mutation causes a ~300-fold reduction, then the mutations increase this ~2-3 fold per round, but the final variant is still ~40-fold less active than the WT? This should be covered more clearly, because it affects some of the claims later on (not necessarily negatively, but it would be easier to follow if this is clear in the next sections).

Yes, good suggestion, we extended the text to give the numbers in the main text and added two tables.

I can't judge the technical aspects of the NMR experiments as I'm not an expert in that area, but I can comment on the general conclusions. For me, the biggest issue is the claim of correlation - but the WT is omitted from most of this analysis for some reason. What would be helpful would be some comparison

between WT and S99T to demonstrate and quantify how the dynamics have changed and how this correlates with the 300-fold loss of activity. For instance

Ln 94-95 the interconversion rate increases from 2.3, to 8.2, to 11.5, which sort of correlates with the increase in k_{cat}/K_M from ~0.4 to ~1.5 to ~2.5. But what is the interconversion rate for WT? Is it 600? It might be in previous papers and was omitted for space issues, but it would really strengthen the conclusions a lot if it can be shown that WT is orders of magnitude faster, like the rate. This would remove any doubt there might be around looking at what are pretty small changes across the S99T>triple mutant trajectory.

The referee is correct: it is the combination between the previously published data and the new data on the mutants obtained by directed evolution that makes the conclusions strong. We have expanded the manuscript in several places to include the comparison with wild-type CypA (previously published), so that all information is in this manuscript.

Ln 99, the correspondence probably isn't remarkable, there are no r^2 values given and it's only three points, it looks excellent for afpf (but the changes are very small) and ok for aapf. But again, why isn't WT included in this, does it also correlate? It might be that there are other factors (in addition to dynamics) that cause the loss of activity in S99T - this would be interesting and wouldn't detract from the overall argument. In fact, it might be surprising if 100% of the changes observed can be attributed solely to these slow dynamics (maybe some mention of this could be made in the discussion).

We have added a paragraph describing the published results on wild-type CypA and why we are convinced that the major contribution in the decrease of k_{cat}/K_M is the interconversion process between the major and minor state. However, the reviewer is correct that there could be other factors involved and that is why the correlation is not perfect. We have added in the figure legend that the fit was performed using "orthogonal distance regression", which takes into account uncertainties in both the x- and y-values, and included the adjusted- R^2 value. It turns out that this parameter is not very meaningful and, therefore, we also report the reduced χ^2 value. The referee is correct that the trend of increased k_{cat}/K_M and faster protein interconversion rates holds true for the wild-type protein. A direct quantitative comparison is, however, not feasible since the populations of the two enzyme forms are inverted in the mutants relative to wild-type CypA, which complicates the comparison between the chymotrypsin assay for wild-type and the mutants. In contrast, the comparison of S99T with the rescue mutants is more direct with only small differences in populations.

In the caption to Figure 4, there is no caption for d, and there is a caption for h, which isn't in the figure.

Thank you for catching this, we apologize for this oversight and have corrected the figure legend.

Ln 128, the Phe113 in conformation is introduced, but it isn't discussed what its relevance is? Is there a functional effect of it being in or out? This section could use a little bit more elaboration.

We have added a sentence and Extended Data Figure 9 to the manuscript about the potential for distinct contacts of the two substrate forms with the two conformations of Phe113:

"... rewired the internal packing to increase the dynamics of Phe113, which likely makes distinct contacts with the cis- and trans- substrate (Extended Data Fig. 9), correlated with surrounding group-I residues during the catalytic cycle."

Ln 183 states experimental evidence of alterations of populations of alternative conformations have been sparse - Campbell et al (NCB 2017) looked at this.

Initially, we only cited the News and Views article that references several papers due to space restrictions. We have now added some of those original references here.

So, overall, expansion of some explanation, a few more refs?, more detail about activities, and inclusion of WT in the correlation analysis/argument, discussion of whether all this change is 100% attributable to dynamics, and some easily digestible figure to quantify loss of dynamics from WT to S99T to the rescued mutant would potentially improve the paper for a general audience. That being said, I do think it is an important piece of work that is suitable for a journal such as this.

I'm happy for this review to be non-anonymous and for the authors to contact me if anything needs to be clarified, Colin Jackson.

Thank you, Colin Jackson, for your enthusiasm and thorough review and insightful tips on how to expand the manuscript to make it easier to read. We have followed all of your suggestions in the revised manuscript and included more references that place this in the context of the field.

Reviewer #3 (Remarks to the Author):

The manuscript by Otten and co-workers presents experimental and computational data that provide evidence for a link between enzyme evolution and protein dynamics. In particular, directed evolution was applied to generate mutants (S99T/C115S and S99T/C115S/I97V) of an impaired variant (S99T) of the human proline isomerase cyclophilin A (CypA) in which catalytic activity is partially restored. NMR and room-temperature (RT) X-ray crystallography were then applied to examine side-chain dynamics and conformational heterogeneity, respectively. The increase in the catalytic rate of the S99T/C115S/I97V mutant is proposed to be due to accelerated interconversion between two functionally essential substates.

The thorough study has been carried out by pioneers and world-experts in the fields of directed enzyme evolution, NMR and RT protein crystallography and data analysis and is a beautiful example of synergistically using NMR and crystallography to elucidate the link between macromolecular structure, dynamics and function. It will be of great interest to the broad readership of Nature Communications. The manuscript would benefit from X-ray crystallographic analysis of S99T/C115S and S99T/I97V mutants (see below) but can be published without these additional data.

We thank the referee for the endorsement of our manuscript, and most importantly very good suggestions to improve the writing of the revised manuscript. We believe that his/her suggestions were very helpful.

Detailed remarks in order of appearance in the text:

Line 64 : ' ... increase in activity ...' it might be adequate mentioning that the catalytic activity of S99T/C115S/I97V is only 3% of that of the WT, thus quantifying the 'partial restoration' mentioned in the abstract.

In response to this comment and those by other reviewers, we have added a paragraph on the comparison of wild-type CypA and the mutant forms. In addition, we also added two tables to the revised manuscript that contain the enzyme kinetic and NMR exchange parameters.

Line 72: '... suggested a direct link between the speed of a conformational change in a dynamic network ...' and what ? Please clarify.

Sorry, we indeed lost three words here, thanks for catching this! We intended to say "... a direct link between catalytic efficiency and the speed....." and have corrected it.

Lines 126-128 : 'Both the C115S and I97V mutations subtly reduce the amino acid size, and combined partially restore the "Phe113-in" conformation as a minor conformation that can now be directly observed in the electron density (Fig. 4b,d-f).'

Without crystallographic analysis of S99T/C115S and S99T/I97V mutants it does not seem justified to state that ' ... both mutations ... combined partially restore the "Phe113-in" conformation as a minor conformation ...'. How can the authors exclude that one of the two mutations is sufficient to occupy the minor Phe113 conformation at a 20% level as is the case in the S99T/C115S/I97V mutant ? If the authors' hypothesis is true, Phe113 should be occupied at less (half ?) than 20% occupancy in both S99T/C115S and S99T/I97V mutants. Maybe such low occupancies would be difficult to determine accurately ? If the authors cannot include X-ray analysis of at least one of the two mutants, they should at least nuance their statement in lines 126 – 128.

We have added and deposited the crystal structure for S99T/C115S (PDB ID: 6BTA). Automated analysis by qFit and subsequent visual inspection did not reveal the alternative conformations. At this resolution, it is difficult to reveal conformations at ~10% occupancy as the reviewer hypothesizes. We have modified the crystallographic table and added a picture of the density to Extended Data Figure 7. We have changed the lines to read:

In S99T/C115S/I97V the Phe113-in to Phe113-out conformations have refined occupancies of approximately 20% and 80%, which is intermediate between the 65% and 35% ratio observed for WT and the 100% occupancy of the Phe113-out conformation observed for S99T and S99T/C115S. While we hypothesize, based on the NMR data, that the Phe113-in conformation is present at low occupancy in both S99T and S99T/C115S, it is not observable in the crystallography data.

Line 128: '... Phe113-in conformation ...' has not been introduced before and its identity and importance only becomes clear when studying fig. 4 in detail. It would already help to name and identify this conformation in the legend of fig. 4.

Thank you for this suggestion, we have now added to the figure legend:

The Phe113-out state corresponds to major WT state and minor S99T/C115S/I97V state. Correspondingly, the Phe113-in state corresponds to the minor WT state and the major S99T/C115S/I97V state.

Line 128: '... minor conformation ...' This statement is only quantified (20% occupancy) in the Methods section. Given the suggested catalytic importance of the minor conformation of Phe113, it might be worth to consider mentioning occupancies of the WT, the S99T and the S99T/C115S/I97V mutants in the main text.

We added the text above, which states the approximate occupancies to clarify this point.

Line 145: 'Why did additional rounds of directed evolution fail to yield further improvement in the catalytic rate?' Even if DE failed, did the authors consider rational approaches? In particular, the referee wonders if avoiding the steric clash between Thr99minor and Val97major (Fig. 4f) in a S99T/C115S/I97A mutant would increase the catalytic rate beyond the one of S99T/C115S/I97V?

Indeed, we have tried this and many other mutations to positions 115 and 97 and have never observed a significant effect. On a broader note, this is an interesting thought and one might speculate why DE did not arrive at mutations that would just do this? As pure speculation (fueled by quite good knowledge about this protein by now...): even smaller residues than Val97 would avoid the steric clash between Thr99minor and Val97major, but also make the active site to "loose" (i.e., increasing the sampling of additional degrees of freedom that are unproductive states). This phenomenon is seen often when making mutations in proteins, and since CypA is tightly packed in the entire dynamic network introducing too much flexibility will likely have a detrimental effect.

Line 193: What is meant by '... long-term goal of rationally improving directed evolution.'? How can 'directed evolution' be 'rationally improved'? Better: 'rationally improving enzymes generated by directed evolution.'?

Line 247: lacks 'respectively'

Legend fig 4: panel (d) is not described. Panel (h) should be panel (g). What is meant by 'major in minor state'?

We thank the reviewer for pointing out the textual inconsistencies and have corrected them.

Extended Data Figure 8: is the gain in conformational heterogeneity of Phe113 in S99T/C115S/I97V wrt S99T significant? If not, please consider removing the time-averaged ensemble refinement.

There is a change in the occupancy of alternative Phe113 states in S99T/C115S/I97V compared to S99T by the ensemble refinement method, which relies on different principles than the qFit multiconformer modeling procedure we used in the main figures. We would like to keep this figure in as it provides an additional validation of the change in conformational heterogeneity across the trajectory.

- 1 Kofron, J. L., Kuzmic, P., Kishore, V., Colon-Bonilla, E. & Rich, D. H. Determination of kinetic constants for peptidyl prolyl cis-trans isomerases by an improved spectrophotometric assay. *Biochemistry* **30**, 6127-6134 (1991).

- 2 Fraser, J. S. *et al.* Hidden alternative structures of proline isomerase essential for catalysis. *Nature* **462**, 669-673, doi:10.1038/nature08615 (2009).
- 3 Eisenmesser, E. Z. *et al.* Intrinsic dynamics of an enzyme underlies catalysis. *Nature* **438**, 117-121, doi:10.1038/nature04105 (2005).
- 4 Vallurupalli, P., Bouvignies, G. & Kay, L. E. Studying "invisible" excited protein states in slow exchange with a major state conformation. *J Am Chem Soc* **134**, 8148-8161, doi:10.1021/ja3001419 (2012).
- 5 Camilloni, C. *et al.* Cyclophilin A catalyzes proline isomerization by an electrostatic handle mechanism. *Proc Natl Acad Sci U S A* **111**, 10203-10208, doi:10.1073/pnas.1404220111 (2014).
- 6 Kern, D., Kern, G., Scherer, G., Fischer, G. & Drakenberg, T. Kinetic analysis of cyclophilin-catalyzed prolyl cis/trans isomerization by dynamic NMR spectroscopy. *Biochemistry* **34**, 13594-13602 (1995).
- 7 Aumuller, T., Jahreis, G., Fischer, G. & Schiene-Fischer, C. Role of prolyl cis/trans isomers in cyclophilin-assisted *Pseudomonas syringae* AvrRpt2 protease activation. *Biochemistry* **49**, 1042-1052, doi:10.1021/bi901813e (2010).

REVIEWERS' COMMENTS:

Reviewer #2 (Remarks to the Author):

I have reviewed the changes and the response to reviewer document and I am satisfied that the paper is improved and suitable for publication.

Reviewer #3 (Remarks to the Author):

In the revised version, the authors have responded adequately to most issues I raised and have modified the manuscript accordingly. Importantly, they crystallized the S99T/C115S mutant and determined its X-ray crystal structure. The analysis of its conformational heterogeneity further corroborates the author's initial conclusion that the C115S and I97V combined partially restore the Phe113-in conformation as a minor conformation.

However, I still feel that the 'rescue' should be quantified in the text to make it clear to the reader that the rescued activity is only a few percent of the WT one, even if this is clear from fig. 1D. I suggest the following change in line 87: '... does only restore X% of the full catalytic efficiency if the wild-type protein (fig. 1d).'

Minor changes:

Line 86: S99T/C115S/I97V > S99T/C115S/I97V

Lines 338 – 340: in the added text, it seems as if Phe113-out and -in have been mixed up. Also, two articles are missing ('... to THE major WT state and THE minor S99T/C115S/I97V state.')

We are delighted that the referees are satisfied with our revisions and are of the opinion that our manuscript is now suitable for publication in Nature Communications. We have addressed their remaining points below and highlighted the changes in the manuscript in orange and using “track changes”.

Reviewer #2

I have reviewed the changes and the response to reviewer document and I am satisfied that the paper is improved and suitable for publication.

We thank reviewer 2 again for his constructive comments and are pleased to see that our revised manuscript addressed all his initial concerns.

Reviewer #3

In the revised version, the authors have responded adequately to most issues I raised and have modified the manuscript accordingly. Importantly, they crystallized the S99T/C115S mutant and determined its X-ray crystal structure. The analysis of its conformational heterogeneity further corroborates the author's initial conclusion that the C115S and I97V combined partially restore the Phe113-in conformation as a minor conformation.

However, I still feel that the ‘rescue’ should be quantified in the text to make it clear to the reader that the rescued activity is only a few percent of the WT one, even if this is clear from fig. 1D. I suggest the following change in line 87: ‘... does only restore X% of the full catalytic efficiency if the wild-type protein (fig. 1d).’

As the reviewer notes, all the data is available in Table 1 and Fig. 1D and from there it is indeed clear that the ‘rescue’ is only a few percent compared to wild-type CypA. We do agree, however, with him/her that stating this in the main text makes it immediately clear to the reader and we have changed the sentence as suggested:

However, directed evolution in the context of the S99T mutation only restores about 2% of the full catalytic efficiency of the wild-type protein (Fig. 1d).

Minor changes:

Line 86: S99T/C115S/I97V > S99T/C115S/I97V

We thank the reviewer for pointing out this textual inconsistency and have corrected it.

Lines 338 – 340: in the added text, it seems as if Phe113-out and –in have been mixed up. Also, two articles are missing (‘... to THE major WT state and THE minor S99T/C115S/I97V state.’)

The reviewer is correct and we apologize for this oversight that only occurred in the legend of Figure 4. We have corrected this and for clarity moved this sentence to the description of panels b and c.

(b) Major and minor side chain conformations are shown for S99T/C115S/I97V (5WC7, 1.43 Å, see Supplementary Table 1) in purple and pink, respectively. The populations are flipped relative to wild-type CypA (3K0N, 1.4 Å)⁸ (c), where major/minor states are shown in red and orange, respectively. The Phe113-in state corresponds to the major WT state and the minor S99T/C115S/I97V state. Correspondingly, the Phe113-out state corresponds to the minor WT state and the major S99T/C115S/I97V state.